# Structural basis of membrane potential coupled vectorial CO$_2$ hydration by the DAB2 complex in chemolithoautotrophs

Yat Kei Lo [1], Michael Seletskiy[1], Stefan Bohn [2], Darja Deobald [3], Timo Glatter [4], Sven T. Stripp [5] & Jan M. Schuller [1,6]

The fixation of dissolved inorganic carbon (DIC) such as CO$_2$ and bicarbonate is fundamental to the global primary production. Many autotrophs depend on a diversity of CO$_2$-concentrating mechanisms (CCMs) to overcome the inefficiency of ribulose-1,5-bisphosphate carboxylase/oxygenase (RuBisCO) and the limited supply of DIC. While cyanobacterial CCMs are well characterized, analogous systems in chemolithoautotrophs, specifically active DIC uptake systems have long been overlooked. Here, we present the cryo-EM analysis of DAB2, an essential membrane protein complex for CO$_2$ uptake in *Halothiobacillus neapolitanus*. The cytoplasmic subunit DabA2 displays a β-carbonic anhydrase-like fold, while the transmembrane subunit DabB2 resembles the proton-conducting subunits of respiratory Complex I. Purified DAB2 binds CO$_2$ independent of protonmotive force (PMF); however, did not spontaneously hydrate CO$_2$. Structural analysis reveals a deeply buried active site only accessible via gated substrate tunnels, suggesting substrate access and catalysis are tightly regulated. A distinct transmembrane helix of DabA2 forms the proton pathway and potentially couples proton translocation to catalysis. These features define a vectorial CO$_2$ hydration mechanism that prohibits reverse bicarbonate dehydration. Our findings establish DAB2 as a prototype of a family of PMF-driven carbonic anhydrases, elucidating a distinct strategy for CO$_2$ capture in non-photosynthetic autotrophs.

The autotrophic carbon fixation by microorganisms is essential for global primary production and forms the basis of food chains in a variety of ecosystems, ranging from oceanic photic zones to hydrothermal vents. A key part of this process is the Calvin–Benson–Bassham (CBB) cycle, in which ribulose-1,5-bisphosphate carboxylase/oxygenase (RuBisCO) catalyzes the incorporation of CO$_2$ into organic molecules. Despite of its importance, RuBisCO has a low turnover rate and is susceptible to competitive inhibition by O$_2$,

which results in a loss of fixed carbon and ATP through photorespiration[1]. Furthermore, in many natural environments, the availability of dissolved inorganic carbon (DIC), the collective pool of aqueous CO$_2$, bicarbonate (HCO$_3^-$) and carbonate (CO$_3^{2-}$), is often restricted due to slow gas exchange, sensitivity to environmental pH and temperature[2,3]. These challenges have driven the widespread evolution of CO$_2$-concentrating mechanisms (CCMs) – protein complexes and microcompartments that accumulate cytoplasmic

[1]Philipps-University Marburg, Department of Chemistry and SYNMIKRO Research Center, Marburg, Germany. [2]Cryo-Electron Microscopy Platform and Institute of Structural Biology, Helmholtz Munich, Neuherberg, Germany. [3]Molecular Environmental Biotechnology, Helmholtz Centre for Environmental Research (UFZ), Leipzig, Germany. [4]Max Planck Institute for Terrestrial Microbiology, Marburg, Germany. [5]Spectroscopy & Biocatalysis, Institute of Chemistry, University of Potsdam, Potsdam, Germany. [6]Microbes-for-Climate (M4C) Cluster of Excellence, Marburg, Germany. ✉e-mail: sven.stripp@uni-potsdam.de; jan.schuller@synmikro.uni-marburg.de

$HCO_3^-$ and elevate the local $CO_2$ concentration around RuBisCO to enhance carbon fixation[4].

Cyanobacteria are the dominant oxygenic phototrophs in many aquatic ecosystems and have developed highly effective CCMs that serve as a benchmark for understanding microbial carbon acquisition[4–6]. These systems rely on two principal components: (I) energy-coupled DIC uptake systems that directly import $HCO_3^-$ or indirectly via hydrating cytoplasmic $CO_2$ to $HCO_3^-$ against the concentration gradient, and (II) proteinaceous microcompartments, known as carboxysomes, which encapsulate RuBisCO alongside carbonic anhydrase (CA), the later providing the substrate for the former based on the DIC pool[7]. A variety of cyanobacterial transporters with different affinities and capacities has been investigated, including the ATP-binding cassette (ABC) type $HCO_3^-$ transporter BCT1[8], the $Na^+$-dependent, high-affinity symporter SbtA[9], and the low-affinity, high-flux BicA transporter[10]. Additionally, some cyanobacteria express vectorial carbonic anhydrases (vCAs) such as the NDH-1MS and NDH-1MS' complexes that function by coupling ferredoxin oxidation to $CO_2$ hydration at the thylakoid membrane[11,12]. Together, these systems enable cyanobacteria to establish a high cytoplasmic $HCO_3^-$ concentration and achieve carboxysomal $CO_2$ levels that are several orders of magnitude higher than the extracellular levels, thereby saturating RuBisCO and minimizing photorespiration[4].

By contrast, active DIC uptake systems of chemolithoautotrophs, organisms that fix $CO_2$ using energy derived from inorganic redox reactions, remain poorly understood. Many such species inhabit environments with low and fluctuating DIC concentrations, including hydrothermal vents and sulfide-rich sediments[2], and are thus expected to have evolved specialized carbon uptake systems. A notable example is the DIC-accumulating complex (DAC) originally identified from deep-sea hydrothermal vent γ-proteobacterium *Thiomicrospira crunogena*[13]. The complex was co-transcribed by two genes (*Tcr_0853* and *Tcr_0854*) which were upregulated in response to DIC scarcity and demonstrated to be essential for growth under DIC-limited condition[14]. Recently, a genome-wide barcoded transposon mutagenesis screen revealed novel variants of DAC in *Halothiobacillus neapolitanus*[15]. This species expressed a two-subunit DAC (DAB2) encoded by *hneap_0211* (DabA2) *and hneap_0212* (DabB2) similar to the Tcr_0853/0854 complex, as well as a three-subunit DAC (DAB1) encoded by *hneap_0907* (DabA1), *hneap_0909* (DabB1), and an additional small hypothetical transmembrane protein (*hneap_0908*). While the cytoplasmic subunits DabA1 and DabA2 were classified as *probable inorganic carbon transporter subunits* (Pfam PF10070; formerly as domain of unknown function DUF2309), DabA2 was predicted to harbor a zinc-dependent active site similar to that found in β-carbonic anhydrase (β-CA) or ζ-carbonic anhydrase (ζ-CA). The transmembrane subunit DabB1 and DabB2 were homologues of NADH:quinone oxidoreductases and Mrp antiporters (Pfam PF00361), suggesting a role in proton or sodium translocation. DabA2 and DabB2 formed a heterodimeric complex that was shown to accumulate intracellular DIC and rescue CA-deficient *E. coli* in a pH-independent manner[15]. Furthermore, in vivo DIC-uptake experiments demonstrated that the complex utilized $CO_2$ rather than $HCO_3^-$ as the main substrate, in which this activity was susceptible to CCCP uncoupling[16]. These observations hinted on the putative role of DAB2 as a proton or sodium motive force coupled vCA.

DACs were phylogenetically widespread and presented in at least 14 bacterial phyla, as well as in the Euryarchaeota[15]. Remarkably, homologues were not restricted to autotrophs but could also be found in heterotrophs and pathogenic species, including *Staphylococcus aureus*, *Vibrio cholerae*, and *Bacillus anthracis*. In contrast to the DAB system, the MpsAB complex, a DAC homologue expressed by *S. aureus*, was proposed to function as a $Na^+/HCO_3^-$ symporter and demonstrated to confer bacterial virulence, in addition to DIC accumulation, possibly in the form of $HCO_3^-$[17,18]. This illustrated the potential mechanistic diversity and physiological repurposing of this protein family.

Despite these functional insights, the molecular basis of DAC activity is still unclear. Key questions include how DACs are structurally organized, how $CO_2$ is accumulated and converted, the source of energy and how it coupled to catalysis, and how the enzyme is regulated to prevent futile cycling. Notably, DAC differ fundamentally from canonical CAs in that their activity is not freely reversible and appears to depend strictly on membrane integrity and electrochemical gradients[15,16].

Here, we present the structural and mechanistic study of the DAB2 complex, a representative of the DAC transporter family underlying chemoautotrophic $CO_2$-concentrating mechanisms. Our cryogenic electron microscopy (cryo-EM) structures revealed a distinctive architecture, possibly coupling proton transfer mediated by DabB2 to $CO_2$ hydration catalyzed by the non-canonical vCA DabA2. Structural comparison with other CAs identified an unconventional active site in DabA2, connected by gated substrate tunnels. Additionally, DabA2 possesses a distinctive transmembrane helical extension that formed the putative proton conduit with DabB2. Fourier-transform infrared (FTIR) spectroscopy showed that the complex lacks catalytic activity in the absence of a membrane potential but exhibits strong $CO_2$ binding. Our findings define the DAB2 complex as a class of CAs driven by protonmotive force (PMF), thereby broadening the mechanistic diversity of bacterial CCMs.

## Results

### Cryo-EM analyses reveal DAB2 in multiple ligand-bound states

To unravel the molecular principle of DAC, we reconstituted the DAB2 complex from *H. neapolitanus* in lipid nanodiscs and determined its structure using cryo-EM single-particle analysis. Our initial attempts to prepare cryo-EM grids with the wild-type DAB2 complex were unsuccessful. Despite using mild detergents and synthetic polymers, the complex was highly unstable and prone to dissociation during sample preparation. To overcome this, we engineered a stabilized single-subunit variant by directly fusing the C-terminus of DabB2 to the N-terminus of DabA2, a construct we termed *Dab2*. This design mirrored the naturally occurring single-subunit DAC system found in *Acidimicrobium ferrooxidans*[16] and had no significant impact on bacterial growth yield (Supplementary Fig. 1a).

We captured *Dab2* in multiple functional states, providing distinct structural insights. By treating the protein with either ~17 mM $CO_2$ or 0.1 M bicarbonate (see Methods), we resolved its $CO_2$-bound (*Dab2-$CO_2$*, 2.8 Å) and bicarbonate-bound (*Dab2-$HCO_3^-$*, 3.2 Å) conformations (Supplementary Fig. 2, 3). Unexpectedly, we also obtained a high-resolution structure of *Dab2* bound to $CO_2$ (*Dab2-ambient*, 2.6 Å) from a separate sample prepared under atmospheric air (< 430 ppm $CO_2$, Supplementary Fig. 4). The exceptional quality of this density map, with a local resolution of 2.3 to 3.0 Å, allowed us to confidently model almost the entire protein, assign non-protein ligands with high accuracy, and identify multiple phospholipids associated with the transmembrane domain. Given the overall best resolution, *Dab2-ambient* served as the reference structure for subsequent analyses unless otherwise specified.

Consistent with previous observations[15], our cryo-EM density maps revealed that *Dab2* assembled as a heterodimer, comprising the cytoplasmic subunit DabA2 positioned directly above the transmembrane subunit DabB2 (Fig. 1a). This structural arrangement was in excellent agreement with the molecular weight of approximately 200 kDa estimated by size-exclusion chromatography (Supplementary Fig. 1d). The overall structures were highly similar among samples prepared under different conditions with a root mean square deviation (RMSD) of 0.27 Å between *Dab2-ambient* and *Dab2-$CO_2$*, as well as 0.47 between *Dab2-ambient* and *Dab2-$HCO_3^-$* (Supplementary Fig. 6a).

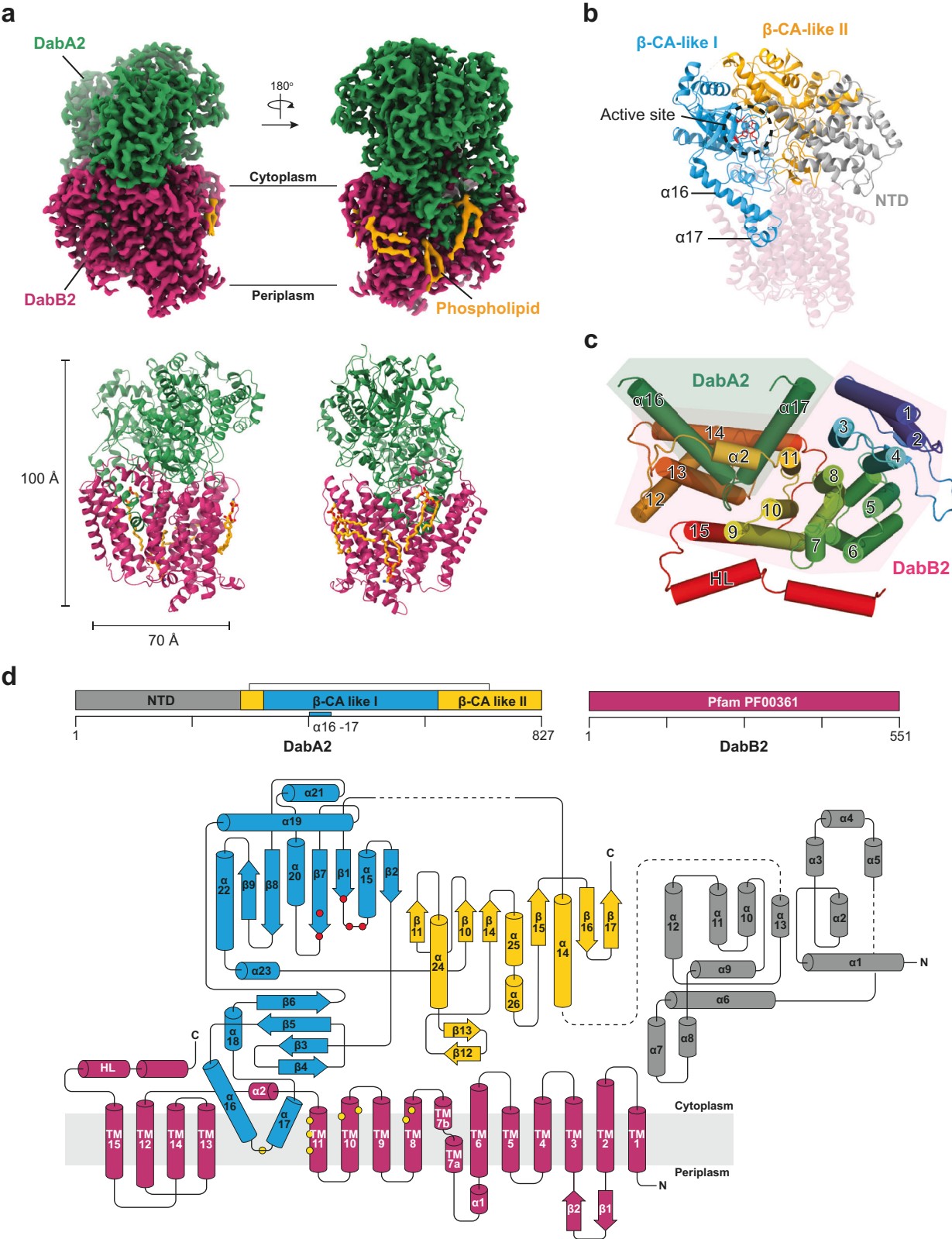

**Fig. 1 | Overall structure of Dab2. a** Cryo-EM density and structural model of the fusion protein complex *Dab2-Ambient* presented in front and back views. DabA2 and DabB2 are colored in green and magenta respectively. The density map revealed several phospholipids (in yellow) interacting with the membrane subunit, DabB2 and DabA2 transmembrane helices. **b** Domain architecture of DabA2. The catalytic core comprised of two β-CA-like domains (in cyan and yellow) and a N-terminal domain (NTD; in gray). Active site residues are highlighted in red. **c** Helices arrangement of DabB2 and DabA2 transmembrane "finger-like" motif (α16, α17) viewed from the cytoplasmic side. HL: lateral helix. **d** Topology of DAB2. Protein domains are colored according to (**b**). Zinc-coordinating residues and putative proton transfer residues are depicted in red and yellow dots respectively. Dash lines indicate unresolved region, likely due to protein flexibility.

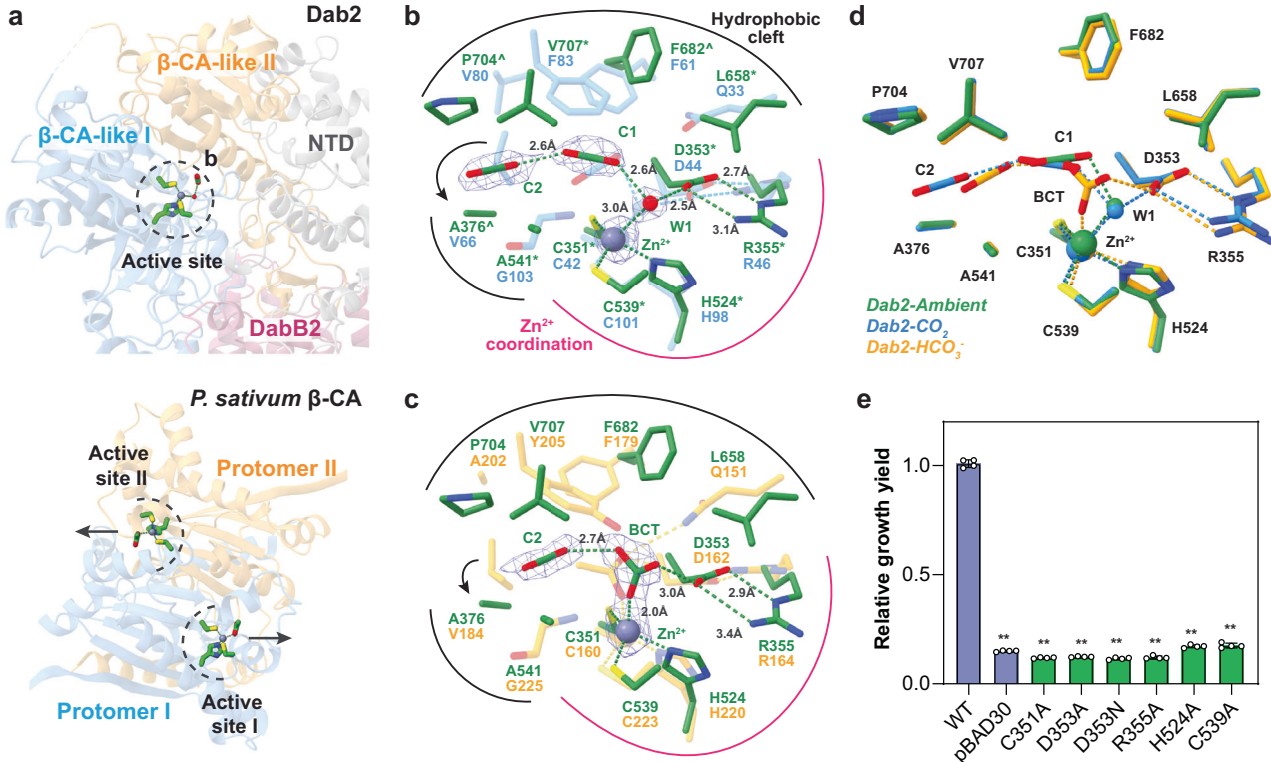

**Fig. 2 | Active site architecture of DabA2 in $CO_2$ and $HCO_3^-$ bound states.**
**a** Comparison of active site location in *Dab2* and *Pisum sativum* β-CA (PSCA; PDB 1EKJ). Arrows indicate immediate connection to the bulk solvent. **b** Superposition of *Dab2-CO2* on *Pseudomonas aeruginosa* β-CA (PACA) in $CO_2$-bound state (PDB 5BQ1; in blue), and **c** *Dab2-HCO3* on *PSCA* in acetate-bound state (in orange). Key molecular interactions are depicted by dashes. Residues strictly conserved or conserved at ≥ 90% sequence identity among DabA2 homologues are marked by asterisk (*) and circumflex (^) respectively. Arrows indicate reposition and substitution of β-CA conserved Val residue (Val66[PACA] and Val184[PSCA]). Notably, the transition state stabilizing residue (Gln33[PACA], Gln151[PSCA]) was substituted by Leu658

in DabA2. Densities of ligands are shown at 9.2 σ (*Dab2-CO2*) and 6.5 σ (*Dab2-HCO3*). **d** Comparison of active site from *Dab2-ambient* (green), *Dab2-HCO3* (yellow), and *Dab2-CO2* (blue) **e** Site-directed mutagenesis of $Zn^{2+}$-coordinating residues. Bar heights and error bars represent means and standard deviations respectively (n = 4 biological replicates). "**" Indicates statistically significant difference compared to WT ($P < 0.05$) according to Holm-Bonferroni corrected two-tailed t-test. *P*-values for pBAD30 = $1.38 \times 10^{-6}$, C351A = $1.38 \times 10^{-6}$, D353A = $1.77 \times 10^{-6}$, D353N = $5.44 \times 10^{-7}$, R355A = $9.74 \times 10^{-8}$, H524A = $1.35 \times 10^{-7}$, C539A = $6.59 \times 10^{-9}$. Bacterial growth yield is presented relative to that of wild-type (WT = 1.0). Source data are provided as a Source Data file.

## DabA2 structurally mimics β-carbonic anhydrases

DabA2 was organized into three distinct domains: a helical N-terminal domain of 280 residues and two β-CA-like domains that form the catalytic core (Fig. 1b, d). Despite the structural resemblance, only five amino acids were strictly conserved between DabA2 and canonical β-CAs (Supplementary Fig. 7). The N-terminal domain comprised twelve intertwined α-helices and lacked any resemblance to other proteins, as confirmed by Foldseek[19] and DALI[20]. Notably, this region was less conserved than the β-CA-like domains among DabA2 homologs (Supplementary Fig. 8), suggesting that its primary role is to maintain DabA2 proper folding rather than to participate directly in catalysis.

β-CAs are typically formed by two symmetrical Rossmann folds and function as a homo-dimer or homo-tetramer[21]. In contrast, the two β-CA-like domains in DabA2 displayed structural adaptations that broke this symmetry. The β-CA-like domain I (residues 341–655) featured an extended "finger-like" amphipathic helix–loop–helix motif (α16 and α17) which inserted deeply into the membrane bilayer, establishing interactions with DabB2 (Fig. 1b–d; Supplementary Fig. 9). Conversely, the β-CA-like domain II formed a discontinuous domain, comprising residues 281–336 and 656–827. It possessed an extended antiparallel β-sheet (β12 and β13), connecting the second and third β-sheets and form the cytoplasmic interaction interface (Fig. 1d). Collectively, this interface, together with the "finger-like" motif, accounts for a buried surface area of 4700 Å² as calculated by the PISA server[22]. A dense network of hydrogen bonds and salt bridges further stabilized the interaction between DabA2 and DabB2, with hydrophobic contacts

predominating between DabA2 helices α16/α17 and DabB2 transmembrane helices 11–14 (Supplementary Table 2, Supplementary Fig. 9). These intricate interfaces may contribute to the protein complex assembly and scaffold key residues for the catalytic function of the DAC transporter system (see below).

## DabA2 binds $CO_2$ and bicarbonate in a distinct active site

DabA2 possessed a putative active site with a region of high electron density located within β-CA-like domain I, positioned near the interface of the two β-CA-like domains (Figs. 1b, 2a, Supplementary Fig. 6b–d). This correlated to the presence of one zinc ion per protein molecule, revealed using inductively coupled plasma mass spectrometry (ICP-MS; Supplementary Fig. 10). The catalytic zinc ion was coordinated by a Cys₂His(H₂O) motif comprised of Cys351, Cys539, and His524 (Fig. 2b), bearing resemblance to that observed in type I β-CAs or the "R-state" of type II β-CAs[21,23]. A zinc-bound water molecule or hydroxide ion was stabilized by the Asp353-Arg355 dyad. In β-CAs, the conserved Asp was proposed to mediate deprotonation of the zinc-bound water, a prerequisite for $HCO_3^-$ formation[24]. When we mutated the zinc-coordinating residues and the Asp-Arg dyad to alanine by site-directed mutagenesis, significantly impaired bacterial growth under $CO_2$-limiting conditions was observed, underscoring their critical functional roles (Fig. 2e). The Ala substitutions had limited effect on the protein expression levels (Supplementary Fig. 11). Together with the structurally resolved water molecules hydrogen-bonded to conserved residues Ser356, His 585, and Asp590, Asp353 might

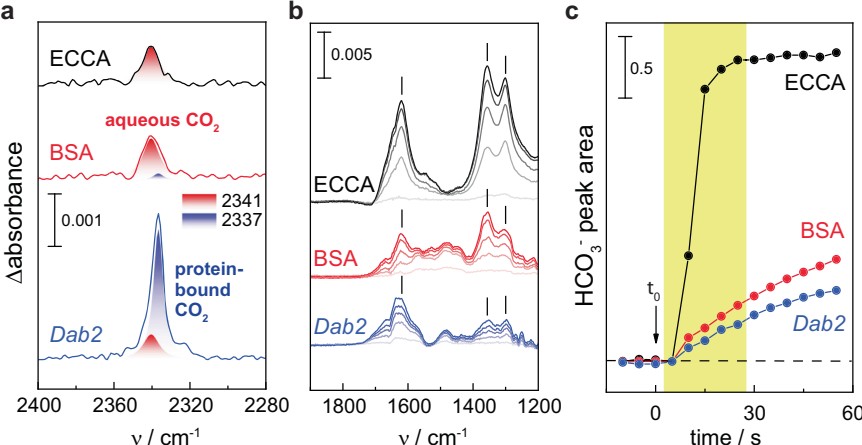

**Fig. 3 | Comparison of CO₂ binding and HCO₃⁻ formation.** Time-resolved "CO₂-minus-N₂" ATR FTIR difference spectra for *E. coli* β-CA (ECCA, black), BSA (red), and *Dab2* (blue). **a** After 25 s in the presence of 10% gaseous CO₂, bands at 2341 and 2337 cm⁻¹ indicate the presence of dissolved CO₂ or protein-bound CO₂, respectively. **b** At lower frequencies, the increasingly more positive bands at 1614, 1360, and 1302 cm⁻¹ revealed HCO₃⁻ formation. All spectra run from light color ($t = 5$ s) to full color ($t = 25$ s). **c** Kinetics of CO₂ hydration with ECCA, BSA, and *Dab2*. The arrow marks the switch from 100% N₂ to 90% N₂ and 10% CO₂($t_0$). Spectra in (**b**) relate to the time frame between 5 and 25 s highlighted in (**c**). Source data are provided as a Source Data file.

serve a similar function in relaying the proton to the bulk solvent in DabA2 (Supplementary Fig. 12).

Our structural analysis further revealed two elongated electron densities near the zinc ion, surrounded by hydrophobic residues in both *Dab2-CO₂* and *Dab2-ambient*, which likely correspond to protein-bound CO₂ molecules (Fig. 2b, Supplementary Fig. 6b, c). Their consistent appearance across independently prepared samples supported the assignment of these densities as specifically bound CO₂. Conventional β-CAs typically accommodate only a single CO₂ or HCO₃⁻ molecule; however, the DabA2 active site appeared to bind two. The first CO₂ molecule (C1) possibly formed a hydrogen bond with the zinc-bound water, suggesting the capture of a substrate-bound state. In β-CAs, the active site is enclosed by a loop and a conserved valine residue, precluding the binding of an additional CO₂ molecule. Yet, in DabA2 the loop between β2 and β3 was displaced away together with the conserved valine replaced by Ala376, thereby expanding the active site cavity (Fig. 2b). This structural rearrangement permitted the accommodation of a second CO₂ molecule (C2), stabilized by hydrophobic interactions within the active site cleft.

In *Dab2-HCO₃*, we identified a prominent triangular electron density adjacent to the zinc ion, likely representing a HCO₃⁻ bound at the active site (Fig. 2c). The structural similarity between *Dab2-HCO₃* and *Dab2-ambient* (RMSD = 0.47 Å), suggests substrates binding had minimal impact on the protein global conformation, as well as the active site architecture (Fig. 2d). While the HCO₃⁻ bound state might be captured during catalytic turnover or due to binding of added ion, we cannot exclude the possibility that this density arises from the non-enzymatic CO₂ hydration due to an increased pH from 7.5 to 7.8 upon addition of NaHCO₃. Even if this was the case, the observed HCO₃⁻ still occupied the active site in a manner comparable to catalytically active β-CAs[25], suggesting a conserved mode of substrate coordination.

To validate the binding of supernumerary CO₂ within the *Dab2* protein complex independent of structural analysis, we employed attenuated total reflectance (ATR) FTIR spectroscopy[26]. This is a robust method for detecting CO₂/HCO₃⁻ conversion and CO₂ protein binding[27–29]. A *Dab2* protein film was formed under inert carrier gas (N₂) before the reaction was started by changing the atmosphere in situ to 10% CO₂, and time-resolved "CO₂-minus-N₂" difference spectra were calculated by subtracting the N₂ background spectrum from all spectra recorded under 10% CO₂ (Fig. 3a, b; see Supplementary Fig. 13a for the complete datasets). Bands at 1614 cm⁻¹, 1360 cm⁻¹, and 1302 cm⁻¹ were assigned to the stretching (ν₂, ν₃) and bending (ν₄)

modes of HCO₃⁻, respectively[29]. These bands appeared positive in the spectra of *E. coli* β-CA (ECCA), equivalent to a fast accumulation of HCO₃⁻ upon enzymatic hydration of CO₂. A feature at higher frequencies was fitted with bands at 2341 and 2337 cm⁻¹, indicative of CO₂ in aqueous solution and CO₂ bound to protein, respectively[27]. This shift relates to differences in proticity and can be mimicked by comparing the IR spectrum of CO₂ in H₂O or DMSO (Supplementary Fig. 14). Overall similar results were observed for BSA, which were utilized as a protein standard to probe "non-catalytic" CO₂ hydration. The HCO₃⁻ formation in BSA was significantly slower than ECCA (Fig. 3b, c), and a similar response was observed for *Dab2*, hinting at non-catalytic CO₂ hydration with both BSA and *Dab2*. In contrast, the CO₂ feature of *Dab2* was dominated by the 2337 cm⁻¹ band, approximately ten times stronger than the aqueous CO₂ signature observed for BSA and ECCA (Fig. 3a). This indicated that membrane potential is not essential for CO₂-binding, even though the protein complex itself did not spontaneously catalyze CO₂ hydration under these conditions.

## DabA2 lacks the canonical residue for stabilizing the transition state

To understand why DabA2 did not readily catalyze CO₂ hydration, we performed a detailed inspection of its active site and compared it with canonical β-CAs. In β-CAs, a conserved glutamine or histidine residue was suggested to stabilize the transition state during catalysis by hydrogen-bonding to the zinc-bound intermediate[25,30], in which substitution of the conserved Gln/His impaired CA activity[31,32]. An exception is observed in *Mycobacterium tuberculosis* β-CA Rv1284, which, despite lacking any equivalent charged or polar residues, retains CA activity. Its high-resolution structure revealed a water molecule occupying the position of the canonical Gln/His sidechain that may serve as an alternative hydrogen bond donor[33]. In stark contrast, DabA2 lacks any analogous residues, as the conserved Gln/His was replaced by a hydrophobic residue (Leu658), which is incapable of fulfilling this role and does not leave room for a water molecule (Fig. 2b). Notably, this substitution was strictly conserved among 150 DabA2 homologues. To assess whether the absence of a canonical hydrogen-bonding residue renders DAB2 dependent on membrane potential, we substituted Leu658 with glutamine to mimic a canonical β-CA. Although this variant supported bacterial growth under CO₂-limiting conditions, it failed to catalyze CO₂ hydration in vitro (Supplementary Fig. 15), implying that catalysis remains contingent upon a membrane potential. Unexpectedly, replacing Leu658 with glycine, alanine, or

asparagine also did not impair DAB2 in vivo activity (Supplementary Fig. 15b). These observations demonstrated that Leu658 was not essential for the function despite being strictly conserved. DabA2 might employ a distinct mechanism for $CO_2$ hydration, independent of the canonical Gln or His residue. Overall, the lack of spontaneous CA activity likely arises from a more intricate structural configuration that extends beyond a mere single residue substitution at the active site.

### The deeply buried active site in DabA2 requires a $CO_2$/$HCO_3^-$ tunnel

The active site of DabA2 was embedded within the protein core, in contrast to the more surface-exposed active sites of conventional β-CAs (Fig. 2a). This structural arrangement mandates a specialized adaptation—a tunnel system that facilitates efficient diffusion of $CO_2$ and $HCO_3^-$ to and from the active site[34,35]. Using CAVER[36], we identified two putative tunnels (T1, T2) extending from the active site in opposite directions, leading to the bulk solvent (Fig. 4a). The first tunnel was lined with hydrophobic residues with a bottleneck radius of 1.3 Å and branched into two entrances (T1a, T1b). This bottleneck was defined by conserved residues Ala376, Phe378, Pro704, and Ala703 (Fig. 4b, c). In addition, tunnels T1b traversed a second bottleneck of similar dimensions, formed by Phe375, His684, Val696, and Ile700, located immediately downstream. Remarkably, density maps of both *Dab2-ambient* and *Dab2-$CO_2$* revealed several probable $CO_2$ molecules (designated C3 to C5) within and near the tunnel, stabilized by hydrophobic residues (Fig. 4a, Supplementary Fig. 16a, b). This observation was in excellent agreement with the IR signature of protein-bound $CO_2$ (Fig. 3a). Although the predicted bottleneck is slightly narrower than the radius of $CO_2$ molecule (~1.6 Å), protein flexibility may permit $CO_2$ diffusion without significant conformational changes.

*Dab2* possesses a second putative tunnel (T2) that connects the active site to the bulk solvent near the DabA2–DabB2 interface, passing through a large cavity (Fig. 4a). Although primarily scaffolded by hydrophobic residues, several water molecules were found in T2 hydrogen-bonded to backbone amides, which could offer a more favorable, polar route for $HCO_3^-$ egress. This tunnel featured a bottleneck radius of 1.1 Å, gated by conserved residues Arg653, Trp656, and Leu629 (Fig. 4b, c). Given both bottlenecks are significantly narrower than the radius of $HCO_3^-$ (~2.3 Å), it suggests that *Dab2* must undergo substantial conformational rearrangements to facilitate product release via either tunnel. Collectively, these observations highlighted the intricate structural adaptations that govern substrate access and product release in DabA2.

### DabB2 and DabA2 form a putative proton conduit

DabB2 was classified as a proton-conducting membrane transporter (Pfam family PF00361) based on sequence similarity[15]. In agreement with this classification, our structure revealed that DabB2 shared a partial structural homology with NuoL, the distal proton-pumping subunit of *E. coli* respiratory Complex I (Fig. 5a, b, Supplementary Fig. 17). Both NuoL and DabB2 features 15 transmembrane helixes, in contrast to 14 in NuoM and NuoN. Moreover, DabB2 possessed a short axial helix (HL), partly resembled that of NuoL (residues 499-527; Figs. 1d, 5a) which is absence in NuoM/N. Notably, DabB2 transmembrane segments TM1–11 aligned closely with those of NuoL, yielding an RMSD as small as 1.05 Å over 235 Cα atoms (residues 82-210, 313-354, 221-238, 251-296). However, subtle differences in helix positioning, combined with the integration of DabA2's extended "finger-like" motif, yielded an overall architecture distinct from NuoL.

NuoL has been proposed to transport protons via two antiparallel helix bundles (TM4–8 and TM9–13) that form an S-shaped, discontinuous water channel lined with polar or charged residues and interspersed by a hydrophobic barrier at the membrane's midplane[37–39]. In DabB2, TM4–11, including the segmented helix TM7a

and TM7b closely resembled NuoL's cytoplasmic half-channel (Fig. 5b). Intriguingly, residues responsible for proton transfer in NuoL were also conserved in DabB2, in which alanine substitutions at these positions significantly impair DAB2 activity (Fig. 5c, d). These observations suggest that DabB2 TM4–11 likely function analogously to NuoL's proton channel.

The major structural divergence was observed in TM12–14. In NuoL, TM12 and TM13 are tightly associated with TM9–11, whereas in DabB2 these helices were tilted outward, repositioning TM14 and TM15 (Fig. 5b). This arrangement exposed the hydrophobic core to the cytoplasm and facilitated interaction with DabA2's "finger-like" motif (Figs. 1d, 5b). A conserved glutamate (Glu444) within this motif took the place of Lys399 and Asp400 in NuoL's broken helix TM12b (Fig. 5c) that forms the periplasmic half-channel entrance[38,40]. Lys399 and Asp400 are involved in the oxidoreductase activity and proton pumping of complex I[41], correspondingly, substituting Glu444 with alanine or glutamine abolished DAB2 activity in vivo (Fig. 5d). This indicates that Glu444 does not merely function as a water-bonding residues but it might directly take part in the proton transfer through protonation and deprotonation. While DabB2 lacked a helix bundle equivalent to TM9–13 in NuoL, the integration of DabA2's "finger-like" motif with DabB2 TM12 and TM13 formed a probable periplasmic half-channel for the proton transfer. This distinctive configuration was highly conserved among DabB proteins and likely constitutes a critical coupling point for mediating DAB2's proton-coupled vCA activity.

Interestingly, DabB2 possessed a structural adaptation that may enhance its proton transfer efficiency. In NuoL, two conserved ion pairs (Lys229/Asp178 and Arg175/Glu144) located near the transmembrane subunits interface have been proposed to alternate between an "Open" and "Close" state conformation in response to electron transfer across the peripheral arm of Complex I, thereby modulate lateral proton transfer between the two half channels[42,43]. While the first ion pair was retained in DabB2 (Lys235/Asp185), the Glu residue necessary for the "Close"-state conformation was substituted by Ile151 (Fig. 5e). This adaptation dissociated the second ion pair and possibly stabilizes DabB2 in the "Open"-state conformation, reducing the energy barrier for proton transfer[43]. To probe the function of these ion pairs, we disrupted their interactions by individually replacing them with alanine by site-directed mutagenesis. All variants could complement CA-deficient *E. coli* strains; however, their growth yield was reduced by 30% to 60% (Fig. 5e). Furthermore, reintroducing a Glu residue in position 151 – which should enable alternation between both conformations, reduced the growth yield by half. These observations supported that the ion pairs modification is required for optimal coupling.

While our structural analysis suggests that DAB2 activity maybe coupled to proton transfer, the homologue MpsAB complex has been argued to harness sodium gradient instead[17,44]. To clarify this ambiguity, we performed a comparable complementation assay as previously reported in sodium transporters-deficient strain (Δ*nhaAB*)[44], expressing DAB2. However, unlike MpsAB, DAB2 could not rescue the mutant strain under sodium stress (Supplementary Fig. 18a). Similarly, DAB2 retained full activity when cells were cultivated in the absence of sodium (Supplementary Fig. 18b). These observations imply sodium is probably not required for DAB2 and that the protein complex appears to transport protons exclusively.

## Discussion

In this study, we present the structural characterization of the DAB2 complex, a representative of membrane potential-dependent $CO_2$ hydration systems found in chemolithoautotrophic bacteria. The structures revealed an architecture that differs mechanistically from the cyanobacterial CCMs characterized to date. This has allowed us to define a family of vectorial carbonic anhydrases (vCAs), which couple unidirectional $CO_2$ hydration to transmembrane proton flux.

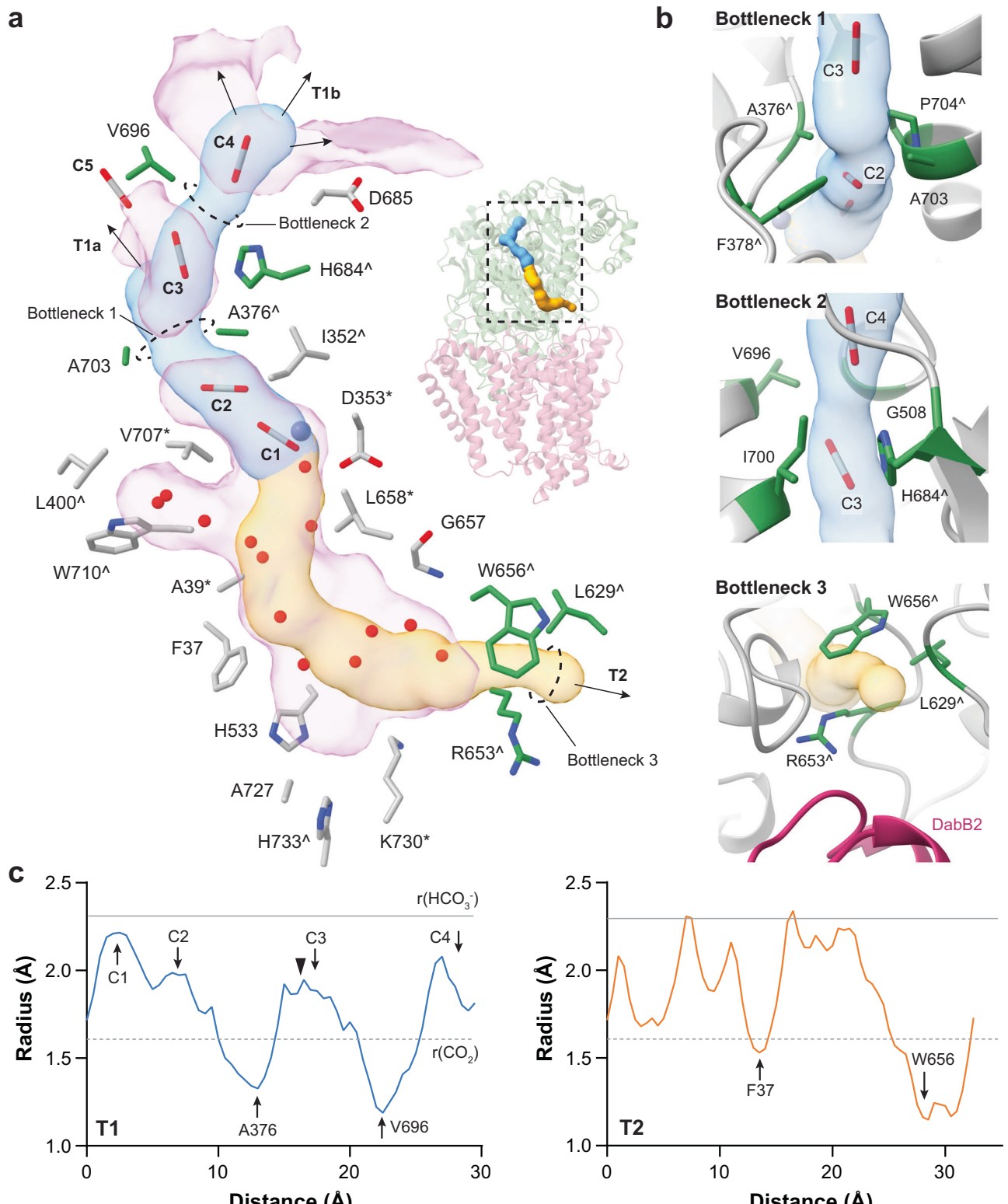

**Fig. 4 | Prediction of tunnels connecting the active site. a** CAVER 3 predicted 2 tunnels (T1, in blue; T2, in orange), connecting the active site to the bulk solvent. T1 was populated with four potential $CO_2$ molecules (C1-C4), and branched into two openings (T1a, T1b). Another potential $CO_2$ molecule (C5) resided near the entrance of T1a. T2 was hydrated by a number of structurally resolved water molecules. Residues lining the tunnel are shown in gray. Bottleneck residues are shown in green. Arrows indicate connection to the bulk solvent. Pink blobs represent cavity calculated by KVFinder[62]. Residues strictly conserved or conserved at ≥ 90% sequence identity among DabA2 homologues are marked by asterisk (*) and circumflex (^) respectively. See Supplementary Fig. 16 for the detailed ligand coordination. **b** Zoomed-in view of the tunnel bottlenecks. **c** Radius along the predicted tunnels, extending from the zinc ion (blue line: T1, orange line: T2). Gray dotted line and solid line indicate the radius of $CO_2$ and $HCO_3^-$, respectively. Arrows mark position of bottlenecks and $CO_2$ molecules. Black triangle indicates the T1a opening. Source data are provided as a Source Data file.

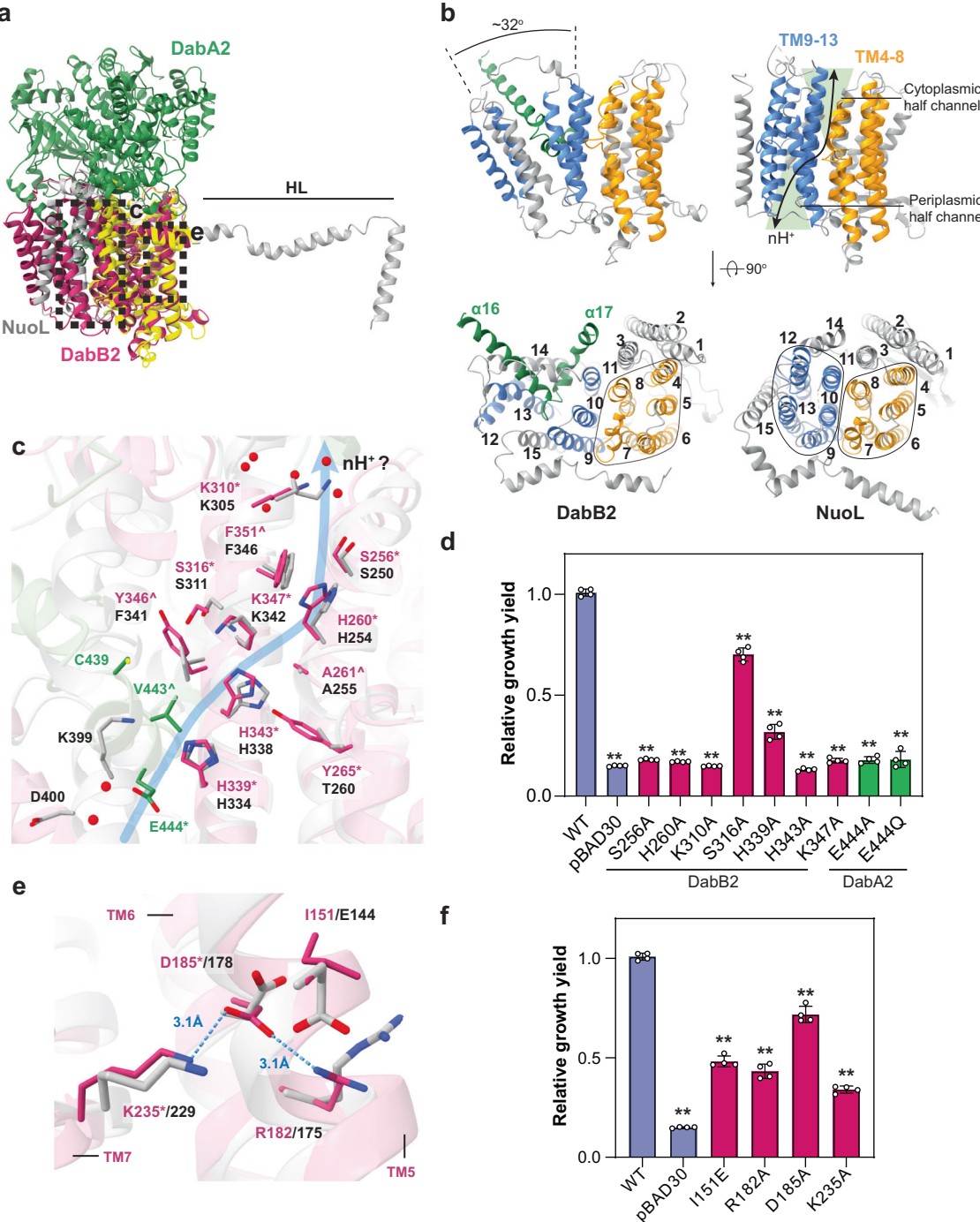

**Fig. 5 | Similarities between DabB2 and NuoL revealed a putative proton pathway. a** Superposition of NuoL (PDB 7P62; in gray) on DabB2. Region aligned to DabB2 is colored in yellow. **b** Topological comparison between DabB2 and NuoL. DabB2 antiparallel helix bundles (in blue and orange) were rearranged to accommodate DabA2 "finger-like" motif (α16, 17; in green). Arrow indicates the proposed NuoL proton transfer pathway. **c** Key residues along the proton pathway were conserved between DabB2 (in magenta) and NuoL (in gray), suggesting *Dab2* might conduct protons in a similar way (depicted by blue line). DabA2 is colored in green. The putative pathway opening was hydrated by several structurally resolved water (red spheres). Residues strictly conserved among DabA2 or DabB2 homologues are marked by asterisk (*). Residues conserved at ≥ 90% sequence identity among homologues are marked by circumflex (^). See Supplementary Fig. 19 for the detailed water coordination. **d** Substitutions of the polar and charged residues

indicated their importance for DAB2 activity. **e** Comparison between DabB2 and NuoL terminal regulatory ion-pairs. Residues of ≥ 80% identity between DabB2 homologues are marked by asterisk (*). Blue dashes depict interactions between DabB2 ion-pairs. **f** Interruptions of these ion-pairs reduced DAB2 activity. **d, f** Bar heights and error bars represent means and standard deviations, respectively ($n = 4$ biological replicates). "**" Indicates statistically significant difference compared to WT ($P < 0.05$) according to Holm-Bonferroni corrected two-tailed t-test. $P$-values for pBAD30 = $1.38 \times 10^{-6}$, S256A = $1.18 \times 10^{-6}$, H260A = $1.82 \times 10^{-6}$, K310A = $1.79 \times 10^{-6}$, S316A = $4.49 \times 10^{-5}$, H339A = $2.28 \times 10^{-6}$, H343A = $1.30 \times 10^{-7}$, K347A = $5.54 \times 10^{-9}$, E444A = $7.32 \times 10^{-10}$, E444Q = $2.01 \times 10^{-6}$, I151E = $3.56 \times 10^{-7}$, R182A = $3.56 \times 10^{-6}$, D185A = $1.76 \times 10^{-4}$, K235A = $2.92 \times 10^{-9}$. Source data are provided as a Source Data file.

Our analysis pinpointed several distinctive features that intricately link CA activity to proton transfer. We obtained structures of DAB2 in lipid nanodiscs in $CO_2$ and $HCO_3^-$-bound states. DabA2 adopts a protein fold homologous to dimeric β-class CA, but its active site architecture diverges substantially. Specifically, the residue that stabilizes the transition state in canonical CAs is replaced by Leu658 in DabA2, which disrupts critical hydrogen bonding and potentially contributes to the enzyme's low basal activity in the absence of proton transfer. Additionally, the active site is buried with the protein core and only accessible via narrow, gated tunnels. This likely impose kinetic barriers to $CO_2$ entry and bicarbonate release. These features suggest a latent catalytic core that is only activated upon proton-driven conformational rearrangement.

DabB2, a membrane-integral subunit, is central to this activation mechanism. It exhibits significant architectural similarity to the antiporter-like subunits of respiratory Complex I, particularly NuoL. Unlike canonical antiporters, however, DabB2 lacks the key ion-pair motifs required for regulating directional proton-pumping, and its structure is modified by the insertion of a long helical extension from DabA2. This "finger-like" transmembrane helix appears to integrate into DabB2, likely contributing to the formation of a periplasmic half-channel for proton conduction. Its placement at the interface of the proton translocation machinery suggests that DabA2 has a dual role as both a structural modulator of proton flux and a sensor of local protonation dynamics. Mutational analysis of conserved residues along the putative proton conduction pathway supports a model in which proton-driven conformational changes initiate catalysis. Together with comparisons to the MpsAB complex, these findings argue against sodium-dependent coupling and instead support PMF-driven activation of DAB2.

Taken together, we hypothesize a PMF-driven vectorial $CO_2$ hydration mechanism (Fig. 6). In the absence of a proton flux, $CO_2$ and water can enter the active site via the substrate tunnel. However, $HCO_3^-$ binding is prevented due to steric hindrance, thus hindering reverse dehydration. A zinc-bound water deprotonates to form a hydroxide ion, with the liberated proton potentially being conducted to the bulk solvent via the conserved residue Asp353. Under PMF-driven conditions, proton transfer through DabB2 leads to the deprotonation or protonation of residues along the conduction pathway. These events possibly modulate the conformation of both the substrate tunnels and the catalytic site through the DabA2 transmembrane "finger-like" motifs, thereby enabling efficient $CO_2$ hydration and $HCO_3^-$ export.

Our findings reveal a highly coordinated regulatory mechanism in which structural rigidity and dynamic tunnel gating converge to control catalysis in the DAC complex. PMF-dependent activation opens the substrate tunnels, enables active site reorganization and permits unidirectional product release. This establishes a structural basis for directionality, while also suppressing the thermodynamically favored reverse reaction of $HCO_3^-$ dehydration. This ensures that catalysis only proceeds under conditions of sufficient PMF, effectively linking carbon uptake to the energetic state of the cell. Unlike ferredoxin-dependent $CO_2$ hydration in cyanobacterial NDH-1 complexes, DAB2 functions independently of electron transfer, representing a distinct adaptation for $CO_2$ capture and conversion in chemolithoautotrophs inhabiting energy- and carbon-limited environments.

## Methods
### Cloning, overexpression and purification of DAB2
The genes *hneap_0212* and *hneap_0211* encoding the DAB2 complex were amplified from *Halothiobacillus neapolitanus* genomic DNA (DSMZ #6118) and cloned into pJS005 (pET-24d modified with a C-terminal 3C protease cleavage site and a twin-strep tag; lab collection) by Gibson Assembly[45]. For the construction of DAB2 fusion (*Dab2*), the stop codon of *hneap_0212* was removed and the gene was directly fused to the start codon of *hneap_0211* as one gene using

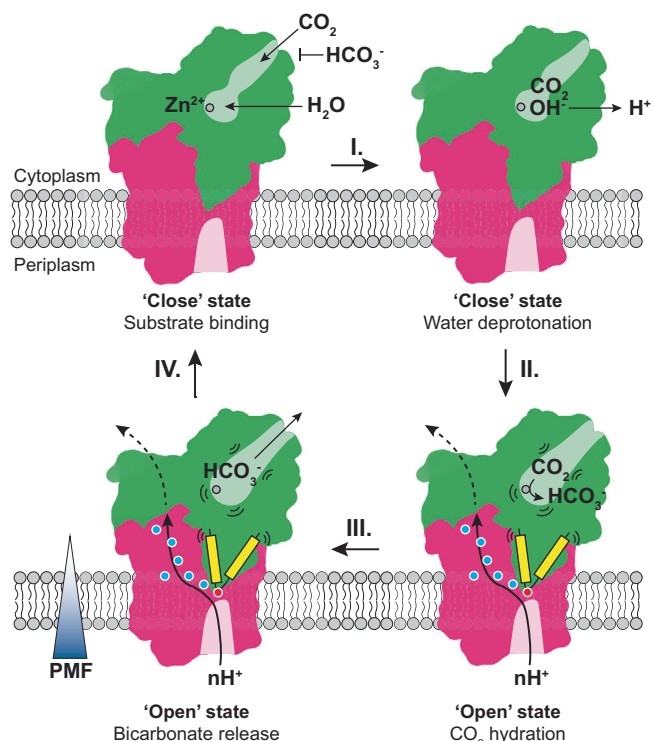

**Fig. 6 | Hypothesized regulatory mechanism.** In the "Close" state, DabA2 binds $CO_2$ and water molecules at the active site, meanwhile $HCO_3^-$ binding is sterically disfavored. The zinc-bound water might deprotonate to form a hydroxide ion (Step **I.**) however, the active site likely remains catalytic inactive at this stage. The protonmotive force (PMF) drives proton transfer across DabB2 via charged and polar residues (blue dots). (De-)Protonation of the DabA2 "finger-like" motifs (yellow) terminal residue Glu444 (red dot) and residues along the proton pathway possibly triggers structural rearrangement at the active and the substrate tunnel, hence activating $CO_2$ hydration (Step **II.**) and subsequent release of bicarbonate formed (Step **III.**). The complex returns to the "Close" state, for substrate binding after releasing the protons and $HCO_3^-$ (Step **IV.**).

commercial KLD reaction mix (NEB #M0554). The plasmids were transformed into chemically competent *E. coli* BL21 (DE3) after verified by sequencing. To generate variants of *Dab2*, specific point mutations were introduced by Golden Gate Assembly following standard protocol[46]. Oligonucleotides (see Supplementary Data 1) and plasmids used in this study are available upon request.

For protein overexpression, bacteria were cultivated at 37 °C in Luria-Bertani (LB) medium supplemented with 25 µg/mL kanamycin and 0.1 mM $ZnSO_4$ until reaching OD600 0.7–0.8. The cultivation temperature was reduced to 16 °C and cells were harvested after 16 h without addition of inducer to reduce protein aggregation.

Cell pellet was resuspended in lysis buffer (20 mM HEPES, 150 mM NaCl, 1 mM PMSF, pH 7.5), and lysed by passing through a microfluidizer (Microfluidics) at 12,000 PSI for three times. Cell debris was removed by centrifugation at 40,000 x g, 4 °C for 20 min, and membrane vesicles were isolated from the supernatant by ultracentrifugation at 200,000 x g, 4 °C for 1 h. The membrane pellet was used immediately or stored at −80 °C until use. To purify DAB2, the pellet was homogenized in lysis buffer at 0.1–0.2 g wet weight per mL buffer. Protein was solubilized with 1% n-dodecyl-β-d-maltoside (DDM) and gently stirred at 4 °C for 1 h. Insoluble material was removed by ultracentrifugation at 200,000 x g, 4 °C for 30 min. The solubilized protein was incubated with pre-equilibrated Strep-Tactin resin (IBA Lifesciences) at 4 °C for 1 hour with gentle agitation. The resin was washed with 15 column volumes of wash buffer (20 mM HEPES, 150 mM NaCl, 0.03% DDM, pH 7.5). The protein was eluted in wash

buffer supplemented with 2.5 mM desthiobiotin and concentrated to 5–7 mg/mL using a 100 kDa cut-off Amicon centrifugal filter (Merck Millipore). The protein was further purified by size-exclusion chromatography (SEC) using a ÄKTA Pure system (Cytiva) and Superose 6 increase 10/200 GL column (Cytiva) in wash buffer. The column was calibrated using the HMW Gel Filtration Calibration Kits (Cytiva) according to the manufacture instruction. Peak fractions were collected and concentrated when necessary. The purity of the proteins was analyzed by SDS-PAGE. Protein was used immediately or supplemented with 10 % glycerol and stored at -80 °C. The same procedure was used to purify Dab2 and mutant variants.

## Nanodisc reconstitution

Membrane scaffold protein MSP1D1 was purified according to established protocol[47]. Chloroform dissolved E. coli polar lipid extract (Avanti Research) was evaporated under a gentle stream of nitrogen to from a thin lipid film. Residue organic solvent was removed under vacuum overnight. Dried lipid was hydrated in reconstitution buffer (20 mM HEPES, 150 mM NaCl, 0.1 M Na-cholate, pH 7.5) to 25 mg/mL and water bath sonicated (Diagenode) at high setting for 30–60 s to aid hydration. 32 uM of Dab2 was incubated with MSP1D1 and lipid at 1:2:40 molar ratio (assuming 800 g/mol for the lipid extract) at 4 °C with gentle agitation. The final reconstitution buffer contained 20 mM HEPES, 150 mM NaCl, 14 mM Na-cholate. A total of 0.25 g pre-equilibrated Bio-beads SM-2 resins (Bio-Rad) was added in two successions – at one hour and 3 hours of incubation. Afterward, the mixture was incubated for 2 more hours. Supernatant was collected and centrifuged at 14,000 x g for 10 min at 4 °C to remove aggregates. The reconstituted protein was analyzed by Superose 6 increase 10/200 GL column in SEC buffer (20 mM HEPES, 150 mM NaCl, pH 7.5). Peak fractions were concentrated when necessary. Sample was immediately used to prepare cryo-EM grids.

## Cryo-EM sample preparation and data collection

For cryo-EM sample preparation, 4 μl of the protein sample (2 mg/mL) was applied to glow-discharged Quantifoil 2/1 Cu200 mesh grids, blotted for 6 s with force 6 in a Vitrobot Mark IV (Thermo Fisher) at 100% humidity and 4 °C, and plunge frozen in liquid ethane. For sample prepared under high $CO_2$ condition ($Dab2\text{-}CO_2$), saturated $CO_2$ water (prepared by bubbling milli-Q water with solid $CO_2$ at 4 °C for at least 30 min, until reaching pH 4.0) was added to the protein sample to final concentration of approximately 17 mM dissolved $CO_2$ shortly before blotting. For sample prepared under high bicarbonate condition ($Dab2\text{-}HCO_3^-$), the protein was incubated in 100 mM $NaHCO_3$ for 2 min prior to blotting. Cryo-EM datasets were collected on a Titan Krios G4i electron microscope operated at 300 kV equipped with a Falcon 4i direct electron detector (Thermofisher scientific). Movies were collected at 165,000x magnification (0.73 Å pixel size) with a defocus range of −0.5 to −2.25 μm.

## Image processing

The processing pipeline for each dataset was summarized in Supplementary Figs. 2–4. All processing steps were carried out in cryoSPARC (v4.7.0)[48]. For Dab2-ambient, 24,092 EER movies were fractionated into 60 fractions per stack without upsampling and motion-corrected using Patch Motion Correction. Contrast transfer function (CTF) was calculated using Patch CTF Estimation. Roughly 200 initial particles were manually picked from a random subset of 100 corrected micrographs as templates to guide optimized blob picking using Blob Picker Tuner. "Junk" particle picks were filtered using Inspect Particle Picks, while the remaining picks were extracted in a 360 pixels box, yielding ~6 million particles and downsampled by bin4, followed by 2D classification. Desirable 2D classes ( ~ 3 million particles) were selected for ab-initio reconstruction in 2 classes. The class containing good particles were refined by Non-uniform Refinement using particles extracted

in full size and 3D classified into 6 classes to further resolve poor particles and heterogeneity. The class with the most prominent $CO_2$-like densities and zinc-bound water was subjected to Heterogenous Refinement in 2 classes. The "best" class was further refined by Non-uniform Refinement and Local Refinement with a mask on the entire protein, excluding the nanodisc density, after particles were polished using Reference-based Motion correction. This resulted in the final reconstruction of 2.64 Å (global FSC = 0.143) with 254,703 particles. A similar strategy was utilized to process the $Dab2\text{-}CO_2$ and $Dab2\text{-}HCO_3^-$ datasets.

For $Dab2\text{-}CO_2$, ~3 million initial particles were extracted from 20,458 micrographs after motion correction, CTF estimation and excluding movies with CTF fits worse than 3.5 Å. Approximately 1.6 million particles were selected after 2D classification to generate 2 classes of an-initio densities. The "good" class was passed to 3D classification in 5 classes and heterogenous refinement in 2 classes. Particles from the 'best' class was polished and used for Non-uniform Refinement followed by Local Refinement, yielding the final reconstruction of 2.72 Å (global FSC = 0.143) with 231,139 particles.

For $Dab2\text{-}HCO_3^-$, ~ 3 million initial particles were extracted from 13,704 micrographs after motion correction, CTF estimation and excluding movies with CTF fits worse than 3.5 Å. After 2D classification, ~2 million particles were selected to generate 2 classes of an-initio densities. The "good" class was 3D classified into 5 classes. The class that contained density corresponding to a zinc-bound bicarbonate was subjected to a round of Non-uniform Refinement. Preferred orientation was mitigated by removing over-populated particles using the Rebalance Orientations tool. The complex was locally refined with a mask on the entire protein using the normalized particle sets, resulting in the final reconstruction of 3.22 Å (global FSC = 0.143) with 226,711 particles.

## Model building and refinement

De novo AlphaFold2 (v1.5.5)[49] predicted model of DAB2 (see Supplementary Data 2) was rigid-body fitted into the densities in ChimeraX (v1.10)[50] and subjected to a round of real-space refinement in Phenix (v1.20.1)[51] to create an initial model. Ligands were modelled manually in COOT (v0.9.8)[52] and the initial model was refined with several rounds of Phenix real-space refinement and manual refinement. Water molecules were added using douse, followed by manual inspections and refinement. The statistics of all cryo-EM data acquisition and refinement are summarized in Supplementary Table 1.

## Site-directed mutagenesis and complementation assay

To investigate the importance of specific amino acid residues, hneap_0212 and hneap_0211 were cloned into pBAD30 which offers a tight genetic regulation for complementation experiments. The sequenced plasmids were transformed into E. coli Lemo21 lacking carbonic anhydrases (ΔcanΔcynT)[14]. This strain could only grow under ambient air if complemented by carbonic anhydrases. Bacteria were cultivated overnight in LB medium supplemented with 25 μg/mL kanamycin under 5 % $CO_2$ atmosphere, then inoculated into fresh LB medium with antibiotic and 0.1% L-arabinose. Growth (absorbance at 600 nm) was monitored using a microplate reader (Tecan) at 37 °C, under ambient atmosphere with agitation. Growth yield was measured at the 10 h time point when the wild-type strain entered stationary phase and the results were presented relative to that of the wild-type.

## Sodium transporter activity assay

Sodium transporter activity was evaluated based on the previously reported method[44]. Genes coding the key sodium transporters (nhaA and nhaB) were deleted by scarless allelic exchange in E. coli MG1655[53]. In brief, 1000 bp flanking sequences upstream and downstream of nhaA and nhaB were cloned into the suicide plasmid pKOV and transformed into E. coli MG1655. Successive homologous

recombinations were performed at 30 °C, followed by plasmid curing at 43 °C and sucrose counter selection. Gene deletion was verified by colony PCR. The double mutant strain was transformed with pBAD30 expressing wild-type DAB2 and routinely cultivated in LB-K (LB with NaCl replaced by KCl). To test the sodium transporter activity, overnight culture was inoculated in LB medium containing 0.1 M NaCl, 25 μg/mL kanamycin and 0.1% L-arabinose. Bacterial growth was monitored as described above.

To verify DAB2 sodium dependency, ΔcanΔcynT E. coli expressing wild-type DAB2 was cultivated in M9 minimal medium[54] prepared with sodium salts (M9-Na) or potassium salts (M9-K), supplemented with 0.4% glycerol. Bacteria were cultivated overnight in M9-K supplemented with 25 μg/mL kanamycin, under 5 % $CO_2$ to deplete cellular sodium prior to inoculation into fresh M9-Na or M9-K. Sodium dependency, as a function of growth was monitored as described above.

### Carbonic anhydrase activity assay

Carbonic anhydrase activity was measured based on the Wilbur-Anderson method[55] using a UV/Vis spectrophotometer (JASCO V-750) at room temperature. 0.3 mL Ice-cold $CO_2$ saturated water was mixed with 0.7 mL assay buffer to initiate $CO_2$ hydration. The reaction mixture contained 5 nM bovine carbonic anhydrase II (Sigma) or 500 nM DDM purified *Dab2*, 20 mM Tris, 150 mM NaCl, 0.03% DDM, 100 μM phenol red, pH 8.3. Acidification as a result of $CO_2$ hydration was monitored at 558 nm absorbance and 0.2 s time-resolution.

### Fourier-transform infrared spectroscopy

All experiments were performed on hydrated protein films in ATR configuration using a FTIR spectrometer (Bruker Tensor27) equipped with a mercury cadmium telluride (MCT) detector cooled by liquid $N_2$. All data were recorded with a spectral resolution of 2 cm$^{-1}$ at 80 kHz scanning velocity. For 25 co-additions of interferometer scans in forward/backward direction, a temporal resolution of 5 s was achieved. The hydration reaction was started by changing the atmosphere from 100% $N_2$ to 90% $N_2$ and 10% $CO_2$ (1 L/min), as reported earlier[29]. For each experiment, 1 μL of 100–200 μM protein solution (ECCA, BSA, *Dab2*) was used and comparable hydration levels were adjusted (Fig. S13) Moreover, all experiments were conducted under ambient temperature (24 °C) and pressure, and in the dark. For the kinetic evaluation, FTIR difference spectra between 1800 and 1200 cm$^{-1}$ were fitted with contributions from $HCO_3^-$ at 1614, 1360, and 1302 cm$^{-1}$ with a FWHM of 53, 48, and 64 cm$^{-1}$, respectively. This allowed calculating the $HCO_3^-$ peak area, which is plotted against time in Fig. 3c as a measure of catalytic activity. Amide band changes at 1650 and 1545 cm$^{-1}$ (FWHM 60 and 55 cm$^{-1}$) are considered in the fit (Fig. S13); this unspecific behavior does not affect the analysis of $HCO_3^-$ kinetics. We used home-written software for fitting, as reported earlier[29].

### Inductively coupled plasma mass spectrometry

For metal ion determination, inductively coupled plasma-triple quadrupole mass spectrometry (ICP-QQQ-MS) was performed. Briefly, purified and desalted protein samples were subjected to acid digestion by incubating them in 11% (v/v) $HNO_3$ (Suprapur grade) for 3 hours at 80 °C. After total hydrolysis, the samples were diluted with ultrapure water to achieve a final $HNO_3$ concentration of 2% (v/v). Calibration standards, ranging from 0.005 μg/L to 500 μg/L, were prepared by serially diluting the ICP multi-element standard solution Merck XVI (Merck Millipore) in 2% (v/v) $HNO_3$. To ensure accuracy, a rhodium internal standard was added to all samples, resulting in a final concentration of 1 μg/L. Metal analysis was conducted using a high-resolution ICP-QQQ-MS system (Agilent 8800, Agilent Technologies) in direct infusion mode with an integrated auto-sampler. The injection system included a Peltier-cooled (2 °C) Scott-type spray chamber equipped with a perfluoroalkoxy alkane (PFA) nebulizer, operating at

0.3 revolutions per second (rps) for 45 s with an internal tube diameter of 1.02 mm. Multiple metals were quantified simultaneously using the Merck XVI standard solution. To reduce polyatomic interferences, the Octopole Reaction System (ORS3) with a collision/reaction cell (CRC) was utilized. Helium (2.5 mL/min) and hydrogen (0.5 mL/min) were introduced into the CRC as collision/reaction gases, while argon was used as the carrier gas at a flow rate of 2.7 mL/min. For each metal, the first (Q1) and second (Q2) quadrupoles were set to the same m/z value, with an integration time of 1 s under auto-detector mode. All measurements were conducted in technical triplicates and normalized using the internal standard, and additional parameters were optimized via the auto-tune function in the MassHunter 4.2 software (Agilent Technologies).

### Sequence conservation analyses

Amino acid conservation analysis was performed using the ConSurf server[56]. Protein sequences of DabA2 and DabB2 were searched against the NCBI non-redundant (nr) protein database, in which 150 sequences (for DabA2) and 100 sequences (for DabB2) were sampled from the list of homologues for the calculation. Structural-based sequence alignment was carried out using the DALI server[20] and visualized in Jalview[57].

### Total proteome analysis

Bacteria were cultivated as described in the complementation assay above (n = 4 biological replicates). The cell pellets were resuspended in 300 μl lysis buffer (2% sodium lauroyl sarcosinate (SLS), 100 mM ammonium bicarbonate) and heated at 90 °C for 40 min. The protein amount was determined by bicinchoninic acid protein assay (Thermo Scientific). Proteins were incubated with 5 mM Tris(2-carboxyethyl) phosphine (Thermo Fischer Scientific) and 10 mM Chloroacetamide at 90 °C for 15 min (Sigma Aldrich) and further processed with SP3 (ref.[58]). Proteins were bound to 4 μl SP3 beads (40% v/v bead stock) in presence of 70% acetonitrile for 15 min at room temperature, followed by two washes of beads with 70% ethanol and an additional wash with acetonitrile. After removal of the supernatant, 1 μg trypsin in 100 mM $NH_4HCO_3$ was added to the beads and digested shaking overnight at 30 °C. Digested proteins were harvested and purified using C18 solid phase extraction. Peptides were finally dried, reconstituted in 0.1% trifluoroacetic acid (TFA) and analyzed using liquid-chromatography-mass spectrometry carried out on an Exploris 480 instrument connected to a VanquishNeo and a nanospray flex ion source (all Thermo Scientific). Peptide separation was performed on a reverse phase HPLC column (75 μm x 26 cm) packed in-house with C18 resin (1.9 μm Reprosil-AQ; Dr. Maisch). The following separating gradient was used: 100% solvent A (0.1% formic acid) to 40% solvent B (99.85% acetonitrile, 0.15% formic acid) over 32 min at a flow rate of 300 nl/min. The direct injection setup was applied.

MS raw data was acquired in data independent acquisition (DIA) mode. The funnel RF level was set to 40. Full MS resolution was set to 120.000 at m/z 200. AGC target value for fragment spectra was set at 3000%. 45 windows of 14 Da were used with an overlap of 1 Da between m/z 320-950. Resolution was set to 15,000 and IT to 22 ms. Stepped HCD collision energy of 25, 27.5, 30 % was used. MS1 data was acquired in profile, MS2 DIA data in centroid mode.

Analysis of DIA data was performed using the DIA-NN version 1.9 (ref.[59]) using a uniprot protein database from *E.coli* BL21 including target proteins to generate a data set specific spectral library for the DIA analysis. The neural network based DIA-NN suite performed noise interference correction (mass correction, RT prediction and precursor/fragment co-elution correlation) and peptide precursor signal extraction of the DIA-NN raw data. The following parameters were used: Full tryptic digest was allowed with two missed cleavage sites, and oxidized methionines (variable) and carbamidomethylated cysteines (fixed). Match between runs and remove likely interferences were enabled. The precursor FDR was set to 1%. The neural network

classifier was set to the single-pass mode, and protein inference was based on genes. Quantification strategy was set to any LC (high accuracy). Cross-run normalization was set to RT-dependent. Library generation was set to smart profiling. DIA-NN outputs were further evaluated using the SafeQuant script[60,61] modified to process DIA-NN outputs

## Reporting summary

Further information on research design is available in the Nature Portfolio Reporting Summary linked to this article.

## Data availability

The atomic models and cryo-EM maps generated in this study have been deposited in the Protein Data Bank (PDB) database and Electron Microscopy Data Bank (EMDB) database, respectively, under accession codes (PDB, EMDB): Dab2-ambient (9RD0, EMDB-53925 [https://www.ebi.ac.uk/emdb/EMD-53925]); Dab2-HCO3- (9RD8, EMDB-53929 [https://www.ebi.ac.uk/emdb/EMD-53929]); Dab2-CO2 (9RD9, EMDB-53930 [https://www.ebi.ac.uk/emdb/EMD-53930]). The atomic models used in this study are available in the PDB database under accession codes: 1EKJ, 5BQ1, 7P62. The mass spectrometry proteomics data have been deposited to the ProteomeXchange Consortium via the PRIDE partner repository with the dataset identifier PXD076189. Oligonucleotides generated in this study and the AlphaFold predicted model of wild-type DAB2 complex are available as Supplementary Data. Source data are provided as a Source Data file. Source data are provided with this paper.

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

## Acknowledgements

J.M.S. acknowledges Prof. Kathleen Scott for kindly providing the carbonic anhydrase deletion strain. Joachim Heberle is acknowledged for providing access to his laboratory and spectrometers at Freie Universität Berlin. Funding: J.M.S. acknowledges funding by the German Research Foundation DFG Emmy Noether grant (SCHaqU 3364/1-1). Sven T. Stripp acknowledges funding by the DFG (STR1554/6-1).

## Author contributions

Y.K.L. and J.M.S. conceived the project. Y.K.L. constructed *E. coli* mutant strains, performed molecular cloning, sequence conservation analysis, biochemical experiments and analyzed the data. Y.K.L. and M.S. performed site-directed mutagenesis, growth assay and analyzed the data. Y.K.L. and S.B. prepared sample for cryo-EM. S.B. collected cryo-EM datasets. Y.K.L. processed the cryo-EM datasets, built, refined, and analyzed the protein structures. S.T.S. performed FTIR experiments and analyzed the data. D.D. conducted ICP-MS experiments and analyzed the data. T.G. performed proteomic experiments and analyzed the data. Y.K.L., S.T.S. and J.M.S. wrote the manuscript with contributions and comments from all authors.

## Funding

## Competing interests

The authors declare no competing interests.
