## [Transparent Peer Review file · Nature Communications]

Structural Basis of Membrane Potential Coupled Vectorial CO₂ Hydration by the DAB2 Complex in Chemolithoautotrophs

Corresponding Author: Dr Jan Schuller

Version 0:

Reviewer comments:

Reviewer #1

(Remarks to the Author)

CO₂-fixation is an essential process for phototrophic and autotrophic organisms. In order to catalyse this process with sufficient efficiency, an increased concentration of physically dissolved CO₂ is usually required. Various CO₂ concentration mechanisms, known as carbon-concentrating mechanisms (CCMs), have evolved over the course of evolution. Of these, the uptake of various forms of CO₂ by transport proteins and the formation of proteinaceous micro compartments has been particularly well studied. In contrast, little is known about the CCMs of chemolithotrophic organisms that live in conditions with rapidly changing CO₂ concentrations. In previous studies, several genes were identified whose products are essential for the survival of *Halothiobacillus neapolitanus* under CO₂ deficiency. Two genes, *DabA2* and *DabB2*, encode a complex, which functions as a putative inorganic carbon transporter. These genes occur in at least 14 bacterial phyla and Euryarchaeota. Lo et al. determined the structure of this complex, describe the effects of defined mutations on growth and present spectroscopic data on CO₂ binding. It is an exciting and very topical subject that suits the journal 'Nature Communications' very well.

In order to obtain a stable preparation, the two subunits of the complex had to be fused together. The authors refer to this construct as *Dab2*. The cryo-EM structures of *Dab2* were obtained in the 'as isolated' state as well as in a 'HCO₃⁻' and a 'CO₂'-bound state. The architecture of the complex, of the carbonic anhydrase (CA) active site and several substrate/product channels are described. Binding of various forms of CO₂ is investigated by IR spectroscopy, key residues are identified by site-directed mutagenesis. The authors develop a model in which CA can bind only CO₂, but not HCO₃⁻, thereby preventing the reverse reaction from occurring. Only protonation of a defined pathway, reminiscent of a similar arrangement in respiratory complex I, leads to a conformational change in a "finger-like" motif. This then enables the chemical reaction and the release of HCO₃⁻.

The manuscript is mainly well written and one can in principle follow the train of thought. Unfortunately, the text is somewhat awkwardly written in places, which hinders the flow of reading in some sections. The quality of the structural data is very good and the description of the active site, the proton pathway and the substrate/product channels can be considered as very reliable to justify the conclusions from this work. Overall, the manuscript contributes new and significant information about CCMs, however, some data requires clarification or more detailed presentation.

One of the authors and other groups has already shown that respiratory complex I of cyanobacteria (Schuller et al., Nat. Commun. 2020) and that of plants and green algae (Klusch et al., Plant Cell 2021) is linked to a CA. It is incorrect to write that these are different things, since in the other cases electron transfer drives CO₂ hydration (l. 494, ff.). This is simply the difference between primary and secondary transport. In cyanobacteria, plants and algae, as described here for the *Dab2* complex, hydration is ultimately driven by proton transfer. The type of CA might differ but the connection to proton transport is clear in all observed cases. This should be taken into account, especially since CA is connected to the homologue of *NuoL* in cyanobacteria and that of *NuoN* in plants and algae.

In this context, it is also unclear to what extent *DabB2* is similar to *NuoL* and not to *NuoM* and *NuoN*. *NuoL*, *M*, and *N* have a highly similar architecture. To what extent is there a specific similarity to *NuoL*? The low RMSD of slightly more than 1 Å over 235 C(alpha) atoms is mentioned (l. 381). It should be specified which region this refers to (amino acid positions) and

how the RMSD in this region compares to NuoM and NuoN. This is important because the two residues Lys399 and Asp400 on NuoL, which are discussed to be functionally relevant in Dab2, are not (or only partially) conserved in NuoM and NuoN.

The discussion of salt bridges in Dab2, which occur in a similar form in NuoL, could be misunderstood (l. 417, ff). In complex I, the opening and closing of the peripheral salt bridge mediates communication between the subunits of the membrane arm. Since Dab2 has only one membrane-bound subunit, this function is not required. It is further discussed in the complex I field that the other salt bridge present in Dab2 ensures a gated proton transfer across the membrane preventing a decoupling of the membrane potential. A similar function can be assumed for Dab2.

The E444X mutations reduce the relative growth yield to the same extent as other mutations in the putative proton channel (Fig. 4d). However, it is argued that this position could therefore not only provide an H-bridge for a water molecule, but could also play a role in proton transfer. However, the effect of the mutation does not differ from that of the other mutations shown in Fig. 4e. On the basis of which data is E444 attributed a special role here?

It is said that the mutations at the salt bridges reduce the activity of the protein by 30% to 60% (l. 424). No activity is measured, the growth rate is diminished.

It should be mentioned again in the legend to Fig. 1 that this is a fusion protein. What is not shown in this figure is that the C-terminus of DabB2 is covalently linked to the N-terminus of DabA2. This means that the position of the NTD in the native complex could also be different. Are there any different topologies proposed by AlphaFold?

The rate of CO₂ hydration by Dab2 compared to that of ECCA and BSA is not clear from Fig. 2e. Firstly, it is not mentioned whether the thick line marks the start or the end of the reaction. Secondly, although the figure clearly shows the different amounts of product formed (intensity), it does not show the kinetics. It would be better to show the time course of the absorbances at 1618, 1358 and 1298 cm⁻¹.

Some minor points:

L. 244, f: Which experiment supports the statement that 'These intricate interfaces are essential for the catalytic function of the DAC transporter system'? Are there site-directed mutations?

The term 'Relative growth yield' (Figs. 2c, 4d, 4f and S13b) should be clarified when used the first time. What is the 100% (or 1.0) value?

The IR frequencies are not consistent between text and figure: 2342 cm⁻¹ (l. 304) and 2341 cm⁻¹ (Fig. 2).

Supplementary Figure 13: The read-out of the photometer starts at around 2.0. Is it certain that this value is still within the linear range of the detector?

L. 416: A 'redox arm' of complex I was never defined, should be replaced by 'peripheral arm'.

L. 670: '... as reported earlier.' Please, cite the report.

L. 671: '... a similar protein concentration () was adjusted.' What was the protein concentration used to generate the hydrated protein film?

Reviewer #2

(Remarks to the Author)

I found this a very interesting, well-written and comprehensive report on the structure and function of the Dab2 complex from *Halothiobacillus neapolitanus*. This system has been identified previously from the Savage lab and is critical to CO₂ concentrating mechanism function in chemoautotrophs. The underlying mechanism of Dab function is critical to our understanding of both how these systems operate, and how we can make use of them in engineered systems. The report is therefore timely and provides important insights into such systems operate. I found the report comprehensive and detailed, leaving very few questions in relation to the science presented. I have a number of minor queries for the authors to consider before acceptance of the manuscript for publication. Otherwise this is an excellent contribution which I support for publication.

The abstract and introduction are well written for authors not commonly publishing in the CCM field. I will note several minor issues that are often found even from experts in the field but would be good to be corrected here. Firstly, preference is to refer to CCMs as 'CO₂ concentrating mechanisms' or 'inorganic carbon concentrating mechanisms' rather than 'carbon'. The latter is a lazy abbreviation of 'inorganic carbon' that moved into the literature some time ago and should be avoided to maintain accurate statements about how these systems operate.

I would argue that chemolithoautotroph CCMs have not been overlooked, per se (line 69). Indeed, the first properly described carboxysomes are those of *Halothiobacillus neapolitanus*. It is perhaps more correct to highlight that we don't know much about the functional uptake of inorganic carbon by these organisms as the first step in their CCM function. Throughout the introduction the authors reasonably use 'DIC' to refer to inorganic carbon species. However, it should be

noted that at times I believe they specifically mean HCO₃⁻ and not CO₂, for example. On line 104, intracellular DIC is almost entirely as HCO₃⁻, held in disequilibrium with CO₂ such that very little CO₂ is present. This is at least the case for cyanobacterial cells and a critical element to CCM function, and it is HCO₃⁻ accumulation that can occur primarily because of its low membrane permeability. CO₂, on the other hand, likely equilibrates with the surrounding environment or is converted to HCO₃⁻ internally by action of Dabs. This is also relevant at line 111, where energy is spent to accumulate CO₂ in the carboxysome, but HCO₃⁻ in the cytoplasm.

Lines 120-123: If possibly provide a reference to support this. In addition, do the authors mean inside or outside the cell when they refer to 'ambient levels'?

Line 125: Again, referring to my earlier comment, we know quite a lot about *Halothiobacillus* carboxysomes and perhaps the distinction here is that we know little about the HCO₃⁻ uptake mechanism(s).

Line 147: Accumulate intracellular DIC or specifically HCO₃⁻/CO₃²⁻?

Line 159: accumulation of charged inorganic carbon species, rather than DIC as this assumes CO₂ is also accumulated... unless there can be a clear distinction here that CO₂ is accumulated in the carboxysome and HCO₃⁻ in the cytoplasm.

In Supplementary Figure 7 there are sequences for Rv1284 and CsoSCA that are not identified in the caption.

I feel that Supplementary Figure 18 should be part of the main text of the manuscript. It is a valuable model that I feel needs to be presented to the general reader, offering a key take-home message.

Reviewer #3

(Remarks to the Author)

Significance: The authors have used cryo-EM to resolve the architecture of the DAB2 complex, providing the first high-resolution structures of a proton-coupled, vectorial carbonic anhydrase representing a surprisingly widespread family of proteins originally described in chemolithotrophs. Distinct from the well-described cyanobacterial CO₂ uptake complexes (e.g., NDH-13/14), the DAB2 structure exhibits a strikingly different configuration: it has a highly modified variant of a canonical β -carbonic anhydrase with a proton-translocating membrane protein homologous to the Mrp/NuoL family of Complex I. This is an exciting discovery since it shows a new type of structural motif for connecting membrane bioenergetics to CO₂ uptake and sets the stage of undoubtedly interesting structure-function studies to understand the mechanism.

General comments:

The authors have successfully developed a heterologous expression strategy to overcome what proved to be the intrinsic instability of the native complex by engineering a stable DabA2–DabB2 fusion, which retained physiological activity and enabled cryo-EM analysis at resolutions of 2.6–3.2 Å. The assignments of the Zn ion, CO₂, and bicarbonate are convincing based upon their analysis. The resulting structures captured multiple substrate-bound states under CO₂, bicarbonate, and ambient air conditions, demonstrating that Dab2 can bind CO₂ molecules even at atmospheric levels. DabA2, while structurally related to β -carbonic anhydrases, displays extensive divergence: it contains two asymmetrical β -CA-like domains, binding two CO₂ molecules simultaneously in its active site. It lacks the canonical H-bonding residue needed to stabilize the catalytic transition state, which the authors attribute as the basis for poor intrinsic hydration activity. The active site lies deeply buried within the protein, that CAVER analysis of their structure shows to be connected to solvent only through narrow tunnels that likely would require conformational changes to permit substrate entry and product release. DabB2, the transmembrane partner, shows close structural homology to the proton-pumping NuoL subunit of Complex I. Functional mutagenesis supports the view that DabB2 forms a proton conducting pathway, coupling substrate handling by DabA2 to proton translocation.

However, the actual mechanism remains largely unexplained, despite experimental tests that have the benefit of eliminating important hypotheses regarding catalytic activity and proton pathways, but not lead to positive assignment regarding how previously described proton transport activity is coupled to CO₂ hydration. Nevertheless, the overall study is a huge advance forward despite the enduring mechanistic mystery.

Specific Comments:

An important limitation of the mutagenesis studies is that the observed phenotypes cannot readily distinguish whether the mutations specifically impair catalytic turnover or proton translocation, as opposed to more general effects on protein stability or complex assembly. Without direct evidence for the proper accumulation of the mutant complexes, the functional consequences remain ambiguous. The application of immunodetection methods—such as antibodies against DabA2 or DabB2 or tags would strengthen the interpretation by confirming that the mutant proteins are expressed and assembled at levels comparable to wild type, thereby ruling out destabilization as the primary explanation for the loss of activity.

The ATR-FTIR experiments provide useful supportive evidence that Dab2 interacts directly with CO₂, consistent with the cryo-EM densities. However, the data do not unambiguously prove supernumerary binding sites, nor do they definitively establish the catalytic vs. non-catalytic nature of hydration. The comparisons to BSA and ECCA are helpful but not definitive controls. The conclusion that Dab2 binds CO₂ "independent of membrane potential" is particularly weak, as the assay lacks physiological coupling conditions. Stronger biochemical validation (e.g., isotope-labeled turnover assays, PMF-dependent reconstitution) will be needed to confirm the functional significance of the FTIR observations.

The hypothesis that the absence of a glutamine or histidine to stabilize the transition state in the DAB protein is a quite reasonable explanation for the absence of catalytic activity in the isolated protein. However, in terms of scholarship, a better citation evidence for the role of the putative transition states stabilizing, glutamine, or histidine that is absent in the DAB active site needs to be presented. The authors need to cite, e.g. a mutational study of that residue rather than the structural

studies if they want to make the argument that this residue is indeed crucial for catalytic activity in the canonical enzymes since the structural studies hypothesize the activity, but do not test it by mutagenesis.

Minor corrections:

Figure 2b: right panel—amino acids are mislabeled

Version 1:

Reviewer comments:

Reviewer #1

(Remarks to the Author)

The authors have taken all of the reviewers' suggestions and comments into account in the revised version of the manuscript. The current version of the manuscript can be accepted, with only a few minor points remaining to be addressed:

L. 75: The term "protonmotive force" is a technical term, whereby the first two words are written together.

L. 162: Delete 'ref.'

L. 197: No activities are shown but growth curves: replace to '... had no significant impact on bacterial growth yield ...'

L. 307: Delete 'ref.'

L. 1208: replace 'explementary' by 'exemplary'

Reviewer #2

(Remarks to the Author)

I appreciate the authors' efforts to address this reviewer's concerns. I am satisfied with the responses to all comments I raised and the changes to the manuscript that have been made.

Reviewer #3

(Remarks to the Author)

The authors have adequately addressed my concerns regarding the manuscript.

REVIEWER COMMENTS

Reviewer #1 (Remarks to the Author):

CO₂-fixation is an essential process for phototrophic and autotrophic organisms. In order to catalyse this process with sufficient efficiency, an increased concentration of physically dissolved CO₂ is usually required. Various CO₂ concentration mechanisms, known as carbon-concentrating mechanisms (CCMs), have evolved over the course of evolution. Of these, the uptake of various forms of CO₂ by transport proteins and the formation of proteinaceous micro compartments has been particularly well studied. In contrast, little is known about the CCMs of chemolithotrophic organisms that live in conditions with rapidly changing CO₂ concentrations. In previous studies, several genes were identified whose products are essential for the survival of *Halothiobacillus neapolitanus* under CO₂ deficiency. Two genes, *DabA2* and *DabB2*, encode a complex, which functions as a putative inorganic carbon transporter. These genes occur in at least 14 bacterial phyla and Euryarchaeota. Lo et al. determined the structure of this complex, describe the effects of defined mutations on growth and present spectroscopic data on CO₂ binding. It is an exciting and very topical subject that suits the journal 'Nature Communications' very well.

In order to obtain a stable preparation, the two subunits of the complex had to be fused together. The authors refer to this construct as *Dab2*. The cryo-EM structures of *Dab2* were obtained in the 'as isolated' state as well as in a 'HCO₃⁻' and a 'CO₂'-bound state. The architecture of the complex, of the carbonic anhydrase (CA) active site and several substrate/product channels are described. Binding of various forms of CO₂ is investigated by IR spectroscopy, key residues are identified by site-directed mutagenesis. The authors develop a model in which CA can bind only CO₂, but not HCO₃⁻, thereby preventing the reverse reaction from occurring. Only protonation of a defined pathway, reminiscent of a similar arrangement in respiratory complex I, leads to a conformational change in a "finger-like" motif. This then enables the chemical reaction and the release of HCO₃⁻.

The manuscript is mainly well written and one can in principle follow the train of thought. Unfortunately, the text is somewhat awkwardly written in places, which hinders the flow of reading in some sections. The quality of the structural data is very good and the description of the active site, the proton pathway and the substrate/product channels can be considered as very reliable to justify the conclusions from this work. Overall, the manuscript contributes new and significant information about CCMs, however, some data requires clarification or more detailed presentation.

One of the authors and other groups has already shown that respiratory complex I of cyanobacteria (Schuller et al., *Nat. Commun.* 2020) and that of plants and green algae (Klusck et al., *Plant Cell* 2021) is linked to a CA. It is incorrect to write that these are different things, since in the other cases electron transfer drives CO₂ hydration (l. 494, ff.). This is simply the difference between primary and secondary transport. In cyanobacteria, plants and algae, as described here for the *Dab2* complex, hydration is ultimately driven by proton transfer. The type of CA might differ but the connection to proton transport is clear in all observed cases. This should be taken into account, especially since CA is connected to the homologue of *NuoL* in cyanobacteria and that of *NuoN* in plants and algae.

Response:

We understand the concern of Reviewer #1. Indeed, DAB2 and the NDH-1MS/MS' complexes of cyanobacteria represent different modes of energy-coupled uptake systems and they are linked to proton transport. However, mechanistically and energetically the NDH-1MS/MS' complexes operate as primary uptake systems, in which electron transfer drives vectorial CO₂ hydration and PMF generation through protons-pumping. In this case, vectorial CO₂ hydration may be catalyzed by protons extrusion to the thylakoid lumen (Schuller et al., Nat. Commun. 2020), therefore energy is derived from electron transfer but not PMF. By contrast, DAB2 functions solely as a secondary uptake system and does not generate PMF, at least when operating as a vectorial carbonic anhydrase. Furthermore, CO₂ hydration is hypothesized to be driven by substrate gating mechanism rather than proton extrusion (see main text). In this sense, as pointed out by Reviewer #1, proton transfer (PMF) is the driving force of CO₂ hydration in DAB2.

In this context, it is also unclear to what extent DabB2 is similar to NuoL and not to NuoM and NuoN. NuoL, M, and N have a highly similar architecture. To what extent is there a specific similarity to NuoL? The low RMSD of slightly more than 1 Å over 235 C(α) atoms is mentioned (l. 381). It should be specified which region this refers to (amino acid positions) and how the RMSD in this region compares to NuoM and NuoN. This is important because the two residues Lys399 and Asp400 on NuoL, which are discussed to be functionally relevant in Dab2, are not (or only partially) conserved in NuoM and NuoN.

Response:

We apologize for the confusion. DabB2 is more similar to NuoL than NuoM/N given their topology and number of transmembrane helices. We have now added a more elaborated description in the main text. We have also included the amino acid positions for the RMSD calculation.

The discussion of salt bridges in Dab2, which occur in a similar form in NuoL, could be misunderstood (l. 417, ff). In complex I, the opening and closing of the peripheral salt bridge mediates communication between the subunits of the membrane arm. Since Dab2 has only one membrane-bound subunit, this function is not required. It is further discussed in the complex I field that the other salt bridge present in Dab2 ensures a gated proton transfer across the membrane preventing a decoupling of the membrane potential. A similar function can be assumed for Dab2.

Response:

Thank you for the comment. As pointed out, Dab2 does not require the ion-pair for communicating with an adjacent transmembrane subunit, however this could not fully explain why the Arg/Glu ion pair is replaced by Arg/Ile. As evidenced by mutagenesis experiment (Fig. 5f), reverting Ile151 to Glu reduced Dab2's activity, as a function of bacterial growth, therefore the substitution might be a specific adaptation for optimizing proton translocation and the coupled catalysis.

The E444X mutations reduce the relative growth yield to the same extent as other mutations in the putative proton channel (Fig. 4d). However, it is argued that this position could therefore

not only provide an H-bridge for a water molecule, but could also play a role in proton transfer. However, the effect of the mutation does not differ from that of the other mutations shown in Fig. 4e. On the basis of which data is E444 attributed a special role here?

Response:

Indeed, substitution of E444 by Ala or Gln reduced the relative growth yield to the same extent. If E444 only provides H-bond to water molecules, the E444Q variant should preserve (partial) proton transfer activity, thus partly complement the growth of CA deficient *E. coli*. However, this is not the case, and only the wild-type (E444) could complement the growth. This implies a titratable residue is required for the proton transfer. In other word, proton transfer from the periplasmic side likely mediates through (de-)protonation of E444, in addition to the water molecule coordinated adjacent to E444.

It is said that the mutations at the salt bridges reduce the activity of the protein by 30% to 60% (l. 424). No activity is measured, the growth rate is diminished.

Response:

Thank you for pointing this out. We have now refined the descriptions.

It should be mentioned again in the legend to Fig. 1 that this is a fusion protein. What is not shown in this figure is that the C-terminus of DabB2 is covalently linked to the N-terminus of DabA2. This means that the position of the NTD in the native complex could also be different. Are there any different topologies proposed by AlphaFold?

Response:

We share the same concern as Reviewer #1 therefore we now included structural comparisons of wild-type DabA2B2 AlphaFold predicted model and the fusion Dab2 protein in Supplementary Figure 1c. There is no significant structural difference between the models. We have also revised the legend of Fig. 1 as suggested.

The rate of CO₂ hydration by Dab2 compared to that of ECCA and BSA is not clear from Fig. 2e. Firstly, it is not mentioned whether the thick line marks the start or the end of the reaction. Secondly, although the figure clearly shows the different amounts of product formed (intensity), it does not show the kinetics. It would be better to show the time course of the absorbances at 1618, 1358 and 1298 cm⁻¹.

Response:

Thank you for the suggestion. The caption of Fig. 3 (previously Fig. 2) now explains that the spectral traces run from “light” to “full” color. Additionally, we included kinetic traces that facilitate comparing the rate of HCO₃⁻ formation for ECCA, BSA, and Dab2 (Fig. 3c). New supporting information, Supplementary Fig. 13b–d explain the process of data fitting that yields the data points in Fig. 3c.

Some minor points:

L. 244, f: Which experiment supports the statement that ‘These intricate interfaces are essential for the catalytic function of the DAC transporter system’? Are there site-directed

mutations?

Response:

We apologize for the confusion. Our structure suggests that the interactions (at the interfaces) likely participate in the assembly of the protein complex. In particular, the key residues E444 of DabA2 not only forms part of the interface between DabB2 and DabA2 helices α 16-17 but essential for the coupled catalysis, as evidenced by site-directed mutagenesis. To clarify, we have revised the main text accordingly.

The term 'Relative growth yield' (Figs. 2c, 4d, 4f and S13b) should be clarified when used the first time. What is the 100% (or 1.0) value?

Response:

The growth yield was presented relative to that of the wild-type strain. The 1.0 value implies an identical growth yield as the wild-type strain. We have now defined 'Relative growth yield' in Fig. 2 legend and Materials and Methods.

The IR frequencies are not consistent between text and figure: 2342 cm⁻¹ (l. 304) and 2341 cm⁻¹ (Fig. 2).

Response:

Thank you for pointing this out. We have now unified the wavenumber.

Supplementary Figure 13: The read-out of the photometer starts at around 2.0. Is it certain that this value is still within the linear range of the detector?

Response:

We appreciate Reviewer #1 careful assessment. According to the manufacturer's specifications, the photometer used in this study (JASCO model V-750) maintains linearity up to 3.0 Abs.

L. 416: A 'redox arm' of complex I was never defined, should be replaced by 'peripheral arm'.

Response:

Thank you for pointing it out. We have revised accordingly.

L. 670: '... as reported earlier.' Please, cite the report.

Response:

We have now included the respective reference.

L. 671: '... a similar protein concentration () was adjusted.' What was the protein concentration used to generate the hydrated protein film?

Response:

We have now added the information (100–200 μM protein). The new spectra in Supplementary Fig. 13d additionally emphasizes that the three protein films (ECCA, BSA, Dab2) are comparable.

Reviewer #2 (Remarks to the Author):

I found this a very interesting, well-written and comprehensive report on the structure and function of the Dab2 complex from *Halothiobacillus neapolitanus*. This system has been identified previously from the Savage lab and is critical to CO₂ concentrating mechanism function in chemoautotrophs. The underlying mechanism of Dab function is critical to our understanding of both how these systems operate, and how we can make use of them in engineered systems. The report is therefore timely and provides important insights into such systems operate. I found the report comprehensive and detailed, leaving very few questions in relation to the science presented. I have a number of minor queries for the authors to consider before acceptance of the manuscript for publication. Otherwise this is an excellent contribution which I support for publication.

The abstract and introduction are well written for authors not commonly publishing in the CCM field. I will note several minor issues that are often found even from experts in the field but would be good to be corrected here. Firstly, preference is to refer to CCMs as 'CO₂ concentrating mechanisms' or 'inorganic carbon concentrating mechanisms' rather than 'carbon'. The latter is a lazy abbreviation of 'inorganic carbon' that moved into the literature some time ago and should be avoided to maintain accurate statements about how these systems operate.

Response:

Thank you for pointing it out. We have clarified CCMs as "CO₂-concentrating mechanisms" in various part of the text.

I would argue that chemolithoautotroph CCMs have not been overlooked, per se (line 69). Indeed, the first properly described carboxysomes are those of *Halothiobacillus neapolitanus*. It is perhaps more correct to highlight that we don't know much about the functional uptake of inorganic carbon by these organisms as the first step in their CCM function.

Response:

As suggested, we have now emphasized the knowledge gap in active DIC uptake systems rather than CCMs in general (see Abstract).

Throughout the introduction the authors reasonably use 'DIC' to refer to inorganic carbon species. However, it should be noted that at times I believe they specifically mean HCO₃⁻ and not CO₂, for example. On line 104, intracellular DIC is almost entirely as HCO₃⁻, held in disequilibrium with CO₂ such that very little CO₂ is present. This is at least the case for cyanobacterial cells and a critical element to CCM function, and it is HCO₃⁻ accumulation that can occur primarily because of its low membrane permeability. CO₂, on the other hand, likely equilibrates with the surrounding environment or is converted to HCO₃⁻ internally by action of Dabs. This is also relevant at line 111, where energy is spent to accumulate CO₂ in the carboxysome, but HCO₃⁻ in the cytoplasm.

Response:

We agree with Reviewer #2. We have now specified "DIC" as cytoplasmic HCO_3^- when appropriate, and clarified the working principles of cyanobacterial CCMs.

Lines 120-123: If possibly provide a reference to support this. In addition, do the authors mean inside or outside the cell when they refer to 'ambient levels'

Response:

Sorry for the confusion. We have now provided a reference and clarified 'ambient levels' as extracellular levels.

Line 125: Again, referring to my earlier comment, we know quite a lot about Halothiobacillus carboxysomes and perhaps the distinction here is that we know little about the HCO_3^- uptake mechanism(s).

Response:

We have now specifically referred the lack of understanding in CCMs to chemoautotrophic active DIC uptake systems.

Line 147: Accumulate intracellular DIC or specifically $\text{HCO}_3^-/\text{CO}_3^{2-}$?

Response:

The cited study investigated DIC uptake as a function of intracellular ^{14}C accumulation, therefore did not distinguish between individual DIC species.

Line 159: accumulation of charged inorganic carbon species, rather than DIC as this assumes CO_2 is also accumulated... unless there can be a clear distinction here that CO_2 is accumulated in the carboxysome and HCO_3^- in the cytoplasm.

Response:

We share the same concern as Reviewer #2. Indeed, bacteria likely accumulate cytoplasmic charged inorganic carbon rather than CO_2 . However, DIC uptake activity of the MpsAB complex has only been demonstrated through the accumulation of intracellular ^{14}C rather than specifically HCO_3^- (see cited studies). Nevertheless, we have added "...in addition to DIC accumulation, possibly in the form of HCO_3^- ..." to point out the probable biochemical function of the Mps system.

In Supplementary Figure 7 there are sequences for Rv1284 and CsoSCA that are not identified in the caption.

Response:

Thank you for pointing it out. We now added the corresponding descriptions in the legend.

I feel that Supplementary Figure 18 should be part of the main text of the manuscript. It is a valuable model that I feel needs to be presented to the general reader, offering a key take-home message.

Response:

As suggested, we have moved Supplementary Figure 18 to the main text (Fig. 6).

Reviewer #3 (Remarks to the Author):

Significance: The authors have used cryo-EM to resolve the architecture of the DAB2 complex, providing the first high-resolution structures of a proton-coupled, vectorial carbonic anhydrase representing a surprisingly widespread family of proteins originally described in chemolithotrophs. Distinct from the well-described cyanobacterial CO₂ uptake complexes (e.g., NDH-13/14), the DAB2 structure exhibits a strikingly different configuration: it has a highly modified variant of a canonical β -carbonic anhydrase with a proton-translocating membrane protein homologous to the Mrp/NuoL family of Complex I. This is an exciting discovery since it shows a new type of structural motif for connecting membrane bioenergetics to CO₂ uptake and sets the stage of undoubtedly interesting structure-function studies to understand the mechanism.

General comments:

The authors have successfully developed a heterologous expression strategy to overcome what proved to be the intrinsic instability of the native complex by engineering a stable DabA2–DabB2 fusion, which retained physiological activity and enabled cryo-EM analysis at resolutions of 2.6–3.2 Å. The assignments of the Zn ion, CO₂, and bicarbonate are convincing based upon their analysis. The resulting structures captured multiple substrate-bound states under CO₂, bicarbonate, and ambient air conditions, demonstrating that Dab2 can bind CO₂ molecules even at atmospheric levels. DabA2, while structurally related to β -carbonic anhydrases, displays extensive divergence: it contains two asymmetrical β -CA-like domains, binding two CO₂ molecules simultaneously in its active site. It lacks the canonical H-bonding residue needed to stabilize the catalytic transition state, which the authors attribute as the basis for poor intrinsic hydration activity. The active site lies deeply buried within the protein, that CAVER analysis of their structure shows to be connected to solvent only through narrow tunnels that likely would require conformational changes to permit substrate entry and product release. DabB2, the transmembrane partner, shows close structural homology to the proton-pumping NuoL subunit of Complex I. Functional mutagenesis supports the view that DabB2 forms a proton conducting pathway, coupling substrate handling by DabA2 to proton translocation.

However, the actual mechanism remains largely unexplained, despite experimental tests that have the benefit of eliminating important hypotheses regarding catalytic activity and proton pathways, but not lead to positive assignment regarding how previously described proton transport activity is coupled to CO₂ hydration. Nevertheless, the overall study is a huge advance forward despite the enduring mechanistic mystery.

Specific Comments:

An important limitation of the mutagenesis studies is that the observed phenotypes cannot readily distinguish whether the mutations specifically impair catalytic turnover or proton translocation, as opposed to more general effects on protein stability or complex assembly. Without direct evidence for the proper accumulation of the mutant complexes, the functional consequences remain ambiguous. The application of immunodetection methods—such as antibodies against DabA2 or DabB2 or tags would strengthen the interpretation by confirming

that the mutant proteins are expressed and assembled at levels comparable to wild type, thereby ruling out destabilization as the primary explanation for the loss of activity.

Response:

We totally agree with Reviewer #3. Indeed, our site-directed mutagenesis experiments only inform the importance of amino acid residues, as reflected by the bacterial growth, rather than their specific function. To probe into the effect of point mutations on protein expression, we now performed additional LC-MS analyses (See Supplementary Fig. 11). Our findings suggest that the amino acid substitutions have minimal impact on the protein expression levels, therefore the observed phenotype is unlikely due to impaired protein expression.

The ATR-FTIR experiments provide useful supportive evidence that Dab2 interacts directly with CO₂, consistent with the cryo-EM densities. However, the data do not unambiguously prove supernumerary binding sites, nor do they definitively establish the catalytic vs. non-catalytic nature of hydration. The comparisons to BSA and ECCA are helpful but not definitive controls. The conclusion that Dab2 binds CO₂ “independent of membrane potential” is particularly weak, as the assay lacks physiological coupling conditions. Stronger biochemical validation (e.g., isotope-labeled turnover assays, PMF-dependent reconstitution) will be needed to confirm the functional significance of the FTIR observations.

Response:

The reviewer is correct, the FTIR data do not allow specifying the CO₂ binding site. However, in combination with the structural data the FTIR spectra provide strong support for supernumerary binding sites: as discussed in the main text, the increased band intensity suggests a high concentration of CO₂. Additionally, the 4–5 cm⁻¹ down-shift indicates an environment with limited access of water, e.g., a hydrophobic protein channel (also see Supplementary Fig. 14).

We agree that a catalytic CO₂ hydration activity of Dab2 cannot be ruled-out completely, based on the FTIR data alone. This is mentioned in the main text now. However, the similarity between Dab2 and BSA clearly points in the direction of “background” or un-catalyzed CO₂ hydration, as it is common in alkaline aqueous solution. We hope that the kinetic comparison in new Fig. 3c (see our comment to reviewer 1) makes this clearer now.

The interactions between CO₂ and purified proteins (as shown by ATR-FTIR experiments) suggests CO₂ binding do not require membrane potential. Nevertheless, it is unclear if PMF modulates the CO₂ binding under physiological conditions. To clarify, we refined our interpretation on line 320: “...This indicated that membrane potential is not essential for CO₂-binding...”. Thank you for suggesting biochemical experiments. Indeed, we are highly interested in the biochemical characterization of Dab2 but unfortunately this is beyond our current capacity. We considered Reviewer #3 suggestions as a probable follow-up study.

The hypothesis that the absence of a glutamine or histidine to stabilize, the transition state in the DAB protein is a quite reasonable explanation for the absence of catalytic activity in the isolated protein. However, in terms of scholarship, a better citation evidence for the role of the punitive transition states stabilizing, glutamine, or histamine that is absent in the DAB active site needs to be presented. The authors need to cite, e.g. a mutational study of that residue

rather than the structural studies if they want to make the argument that this residue is indeed crucial for catalytic activity in the canonical enzymes since the structural studies hypothesize the activity, but do not test it by mutagenesis.

Response:

Thank you for pointing this out. We have now included relevant references (ref. 31, 32).

Minor corrections:

Figure 2b: right panel—amino acids are mislabeled

Response:

We apologize for the mistake. We have now corrected the AA labels.

REVIEWER COMMENTS

Reviewer #1 (Remarks to the Author):

CO₂-fixation is an essential process for phototrophic and autotrophic organisms. In order to catalyse this process with sufficient efficiency, an increased concentration of physically dissolved CO₂ is usually required. Various CO₂ concentration mechanisms, known as carbon-concentrating mechanisms (CCMs), have evolved over the course of evolution. Of these, the uptake of various forms of CO₂ by transport proteins and the formation of proteinaceous micro compartments has been particularly well studied. In contrast, little is known about the CCMs of chemolithotrophic organisms that live in conditions with rapidly changing CO₂ concentrations. In previous studies, several genes were identified whose products are essential for the survival of *Halothiobacillus neapolitanus* under CO₂ deficiency. Two genes, *DabA2* and *DabB2*, encode a complex, which functions as a putative inorganic carbon transporter. These genes occur in at least 14 bacterial phyla and Euryarchaeota. Lo et al. determined the structure of this complex, describe the effects of defined mutations on growth and present spectroscopic data on CO₂ binding. It is an exciting and very topical subject that suits the journal 'Nature Communications' very well.

In order to obtain a stable preparation, the two subunits of the complex had to be fused together. The authors refer to this construct as Dab2. The cryo-EM structures of Dab2 were obtained in the 'as isolated' state as well as in a 'HCO₃⁻' and a 'CO₂' -bound state. The architecture of the complex, of the carbonic anhydrase (CA) active site and several substrate/product channels are described. Binding of various forms of CO₂ is investigated by IR spectroscopy, key residues are identified by site-directed mutagenesis. The authors develop a model in which CA can bind only CO₂, but not HCO₃⁻, thereby preventing the reverse reaction from occurring. Only protonation of a defined pathway, reminiscent of a similar arrangement in respiratory complex I, leads to a conformational change in a "finger-like" motif. This then enables the chemical reaction and the release of HCO₃⁻.

The manuscript is mainly well written and one can in principle follow the train of thought. Unfortunately, the text is somewhat awkwardly written in places, which hinders the flow of reading in some sections. The quality of the structural data is very good and the description of the active site, the proton pathway and the substrate/product channels can be considered as very reliable to justify the conclusions from this work. Overall, the manuscript contributes new and significant information about CCMs, however, some data requires clarification or more detailed presentation.

One of the authors and other groups has already shown that respiratory complex I of cyanobacteria (Schuller et al., *Nat. Commun.* 2020) and that of plants and green algae (Klusck et al., *Plant Cell* 2021) is linked to a CA. It is incorrect to write that these are different things, since in the other cases electron transfer drives CO₂ hydration (l. 494, ff.). This is simply the difference between primary and secondary transport. In cyanobacteria, plants and algae, as described here for the Dab2 complex, hydration is ultimately driven by proton transfer. The type of CA might differ but the connection to proton transport is clear in all observed cases. This should be taken into account, especially since CA is connected to the homologue of NuoL in cyanobacteria and that of NuoN in plants and algae.

Response:

We understand the concern of Reviewer #1. Indeed, DAB2 and the NDH-1MS/MS' complexes of cyanobacteria represent different modes of energy-coupled uptake systems and they are linked to proton transport. However, mechanistically and energetically the NDH-1MS/MS' complexes operate as primary uptake systems, in which electron transfer drives vectorial CO₂ hydration and PMF generation through protons-pumping. In this case, vectorial CO₂ hydration may be catalyzed by protons extrusion to the thylakoid lumen (Schuller et al., Nat. Commun. 2020), therefore energy is derived from electron transfer but not PMF. By contrast, DAB2 functions solely as a secondary uptake system and does not generate PMF, at least when operating as a vectorial carbonic anhydrase. Furthermore, CO₂ hydration is hypothesized to be driven by substrate gating mechanism rather than proton extrusion (see main text). In this sense, as pointed out by Reviewer #1, proton transfer (PMF) is the driving force of CO₂ hydration in DAB2.

In this context, it is also unclear to what extent DabB2 is similar to NuoL and not to NuoM and NuoN. NuoL, M, and N have a highly similar architecture. To what extent is there a specific similarity to NuoL? The low RMSD of slightly more than 1 Å over 235 C(alpha) atoms is mentioned (l. 381). It should be specified which region this refers to (amino acid positions) and how the RMSD in this region compares to NuoM and NuoN. This is important because the two residues Lys399 and Asp400 on NuoL, which are discussed to be functionally relevant in Dab2, are not (or only partially) conserved in NuoM and NuoN.

Response:

We apologize for the confusion. DabB2 is more similar to NuoL than NuoM/N given their topology and number of transmembrane helices. We have now added a more elaborated description in the main text. We have also included the amino acid positions for the RMSD calculation.

The discussion of salt bridges in Dab2, which occur in a similar form in NuoL, could be misunderstood (l. 417, ff). In complex I, the opening and closing of the peripheral salt bridge mediates communication between the subunits of the membrane arm. Since Dab2 has only one membrane-bound subunit, this function is not required. It is further discussed in the complex I field that the other salt bridge present in Dab2 ensures a gated proton transfer across the membrane preventing a decoupling of the membrane potential. A similar function can be assumed for Dab2.

Response:

Thank you for the comment. As pointed out, Dab2 does not require the ion-pair for communicating with an adjacent transmembrane subunit, however this could not fully explain why the Arg/Glu ion pair is replaced by Arg/Ile. As evidenced by mutagenesis experiment (Fig. 5f), reverting Ile151 to Glu reduced Dab2's activity, as a function of bacterial growth, therefore the substitution might be a specific adaptation for optimizing proton translocation and the coupled catalysis.

The E444X mutations reduce the relative growth yield to the same extent as other mutations in the putative proton channel (Fig. 4d). However, it is argued that this position could therefore

not only provide an H-bridge for a water molecule, but could also play a role in proton transfer. However, the effect of the mutation does not differ from that of the other mutations shown in Fig. 4e. On the basis of which data is E444 attributed a special role here?

Response:

Indeed, substitution of E444 by Ala or Gln reduced the relative growth yield to the same extent. If E444 only provides H-bond to water molecules, the E444Q variant should preserve (partial) proton transfer activity, thus partly complement the growth of CA deficient *E. coli*. However, this is not the case, and only the wild-type (E444) could complement the growth. This implies a titratable residue is required for the proton transfer. In other word, proton transfer from the periplasmic side likely mediates through (de-)protonation of E444, in addition to the water molecule coordinated adjacent to E444.

It is said that the mutations at the salt bridges reduce the activity of the protein by 30% to 60% (l. 424). No activity is measured, the growth rate is diminished.

Response:

Thank you for pointing this out. We have now refined the descriptions.

It should be mentioned again in the legend to Fig. 1 that this is a fusion protein. What is not shown in this figure is that the C-terminus of DabB2 is covalently linked to the N-terminus of DabA2. This means that the position of the NTD in the native complex could also be different. Are there any different topologies proposed by AlphaFold?

Response:

We share the same concern as Reviewer #1 therefore we now included structural comparisons of wild-type DabA2B2 AlphaFold predicted model and the fusion Dab2 protein in Supplementary Figure 1c. There is no significant structural difference between the models. We have also revised the legend of Fig. 1 as suggested.

The rate of CO₂ hydration by Dab2 compared to that of ECCA and BSA is not clear from Fig. 2e. Firstly, it is not mentioned whether the thick line marks the start or the end of the reaction. Secondly, although the figure clearly shows the different amounts of product formed (intensity), it does not show the kinetics. It would be better to show the time course of the absorbances at 1618, 1358 and 1298 cm⁻¹.

Response:

Thank you for the suggestion. The caption of Fig. 3 (previously Fig. 2) now explains that the spectral traces run from “light” to “full” color. Additionally, we included kinetic traces that facilitate comparing the rate of HCO₃⁻ formation for ECCA, BSA, and Dab2 (Fig. 3c). New supporting information, Supplementary Fig. 13b–d explain the process of data fitting that yields the data points in Fig. 3c.

Some minor points:

L. 244, f: Which experiment supports the statement that ‘These intricate interfaces are essential for the catalytic function of the DAC transporter system’? Are there site-directed

mutations?

Response:

We apologize for the confusion. Our structure suggests that the interactions (at the interfaces) likely participate in the assembly of the protein complex. In particular, the key residues E444 of DabA2 not only forms part of the interface between DabB2 and DabA2 helices α 16-17 but essential for the coupled catalysis, as evidenced by site-directed mutagenesis. To clarify, we have revised the main text accordingly.

The term 'Relative growth yield' (Figs. 2c, 4d, 4f and S13b) should be clarified when used the first time. What is the 100% (or 1.0) value?

Response:

The growth yield was presented relative to that of the wild-type strain. The 1.0 value implies an identical growth yield as the wild-type strain. We have now defined 'Relative growth yield' in Fig. 2 legend and Materials and Methods.

The IR frequencies are not consistent between text and figure: 2342 cm⁻¹ (l. 304) and 2341 cm⁻¹ (Fig. 2).

Response:

Thank you for pointing this out. We have now unified the wavenumber.

Supplementary Figure 13: The read-out of the photometer starts at around 2.0. Is it certain that this value is still within the linear range of the detector?

Response:

We appreciate Reviewer #1 careful assessment. According to the manufacturer's specifications, the photometer used in this study (JASCO model V-750) maintains linearity up to 3.0 Abs.

L. 416: A 'redox arm' of complex I was never defined, should be replaced by 'peripheral arm'.

Response:

Thank you for pointing it out. We have revised accordingly.

L. 670: '... as reported earlier.' Please, cite the report.

Response:

We have now included the respective reference.

L. 671: '... a similar protein concentration () was adjusted.' What was the protein concentration used to generate the hydrated protein film?

Response:

We have now added the information (100–200 μM protein). The new spectra in Supplementary Fig. 13d additionally emphasizes that the three protein films (ECCA, BSA, Dab2) are comparable.

Reviewer #2 (Remarks to the Author):

I found this a very interesting, well-written and comprehensive report on the structure and function of the Dab2 complex from *Halothiobacillus neapolitanus*. This system has been identified previously from the Savage lab and is critical to CO₂ concentrating mechanism function in chemoautotrophs. The underlying mechanism of Dab function is critical to our understanding of both how these systems operate, and how we can make use of them in engineered systems. The report is therefore timely and provides important insights into such systems operate. I found the report comprehensive and detailed, leaving very few questions in relation to the science presented. I have a number of minor queries for the authors to consider before acceptance of the manuscript for publication. Otherwise this is an excellent contribution which I support for publication.

The abstract and introduction are well written for authors not commonly publishing in the CCM field. I will note several minor issues that are often found even from experts in the field but would be good to be corrected here. Firstly, preference is to refer to CCMs as 'CO₂ concentrating mechanisms' or 'inorganic carbon concentrating mechanisms' rather than 'carbon'. The latter is a lazy abbreviation of 'inorganic carbon' that moved into the literature some time ago and should be avoided to maintain accurate statements about how these systems operate.

Response:

Thank you for pointing it out. We have clarified CCMs as "CO₂-concentrating mechanisms" in various part of the text.

I would argue that chemolithoautotroph CCMs have not been overlooked, per se (line 69). Indeed, the first properly described carboxysomes are those of *Halothiobacillus neapolitanus*. It is perhaps more correct to highlight that we don't know much about the functional uptake of inorganic carbon by these organisms as the first step in their CCM function.

Response:

As suggested, we have now emphasized the knowledge gap in active DIC uptake systems rather than CCMs in general (see Abstract).

Throughout the introduction the authors reasonably use 'DIC' to refer to inorganic carbon species. However, it should be noted that at times I believe they specifically mean HCO₃⁻ and not CO₂, for example. On line 104, intracellular DIC is almost entirely as HCO₃⁻, held in disequilibrium with CO₂ such that very little CO₂ is present. This is at least the case for cyanobacterial cells and a critical element to CCM function, and it is HCO₃⁻ accumulation that can occur primarily because of its low membrane permeability. CO₂, on the other hand, likely equilibrates with the surrounding environment or is converted to HCO₃⁻ internally by action of Dabs. This is also relevant at line 111, where energy is spent to accumulate CO₂ in the carboxysome, but HCO₃⁻ in the cytoplasm.

Response:

We agree with Reviewer #2. We have now specified "DIC" as cytoplasmic HCO_3^- when appropriate, and clarified the working principles of cyanobacterial CCMs.

Lines 120-123: If possibly provide a reference to support this. In addition, do the authors mean inside or outside the cell when they refer to 'ambient levels'

Response:

Sorry for the confusion. We have now provided a reference and clarified 'ambient levels' as extracellular levels.

Line 125: Again, referring to my earlier comment, we know quite a lot about Halothiobacillus carboxysomes and perhaps the distinction here is that we know little about the HCO_3^- uptake mechanism(s).

Response:

We have now specifically referred the lack of understanding in CCMs to chemoautotrophic active DIC uptake systems.

Line 147: Accumulate intracellular DIC or specifically $\text{HCO}_3^-/\text{CO}_3^{2-}$?

Response:

The cited study investigated DIC uptake as a function of intracellular ^{14}C accumulation, therefore did not distinguish between individual DIC species.

Line 159: accumulation of charged inorganic carbon species, rather than DIC as this assumes CO_2 is also accumulated... unless there can be a clear distinction here that CO_2 is accumulated in the carboxysome and HCO_3^- in the cytoplasm.

Response:

We share the same concern as Reviewer #2. Indeed, bacteria likely accumulate cytoplasmic charged inorganic carbon rather than CO_2 . However, DIC uptake activity of the MpsAB complex has only been demonstrated through the accumulation of intracellular ^{14}C rather than specifically HCO_3^- (see cited studies). Nevertheless, we have added "...in addition to DIC accumulation, possibly in the form of HCO_3^- ..." to point out the probable biochemical function of the Mps system.

In Supplementary Figure 7 there are sequences for Rv1284 and CsoSCA that are not identified in the caption.

Response:

Thank you for pointing it out. We now added the corresponding descriptions in the legend.

I feel that Supplementary Figure 18 should be part of the main text of the manuscript. It is a valuable model that I feel needs to be presented to the general reader, offering a key take-home message.

Response:

As suggested, we have moved Supplementary Figure 18 to the main text (Fig. 6).

Reviewer #3 (Remarks to the Author):

Significance: The authors have used cryo-EM to resolve the architecture of the DAB2 complex, providing the first high-resolution structures of a proton-coupled, vectorial carbonic anhydrase representing a surprisingly widespread family of proteins originally described in chemolithotrophs. Distinct from the well-described cyanobacterial CO₂ uptake complexes (e.g., NDH-13/14), the DAB2 structure exhibits a strikingly different configuration: it has a highly modified variant of a canonical β -carbonic anhydrase with a proton-translocating membrane protein homologous to the Mrp/NuoL family of Complex I. This is an exciting discovery since it shows a new type of structural motif for connecting membrane bioenergetics to CO₂ uptake and sets the stage of undoubtedly interesting structure-function studies to understand the mechanism.

General comments:

The authors have successfully developed a heterologous expression strategy to overcome what proved to be the intrinsic instability of the native complex by engineering a stable DabA2–DabB2 fusion, which retained physiological activity and enabled cryo-EM analysis at resolutions of 2.6–3.2 Å. The assignments of the Zn ion, CO₂, and bicarbonate are convincing based upon their analysis. The resulting structures captured multiple substrate-bound states under CO₂, bicarbonate, and ambient air conditions, demonstrating that Dab2 can bind CO₂ molecules even at atmospheric levels. DabA2, while structurally related to β -carbonic anhydrases, displays extensive divergence: it contains two asymmetrical β -CA-like domains, binding two CO₂ molecules simultaneously in its active site. It lacks the canonical H-bonding residue needed to stabilize the catalytic transition state, which the authors attribute as the basis for poor intrinsic hydration activity. The active site lies deeply buried within the protein, that CAVER analysis of their structure shows to be connected to solvent only through narrow tunnels that likely would require conformational changes to permit substrate entry and product release. DabB2, the transmembrane partner, shows close structural homology to the proton-pumping NuoL subunit of Complex I. Functional mutagenesis supports the view that DabB2 forms a proton conducting pathway, coupling substrate handling by DabA2 to proton translocation.

However, the actual mechanism remains largely unexplained, despite experimental tests that have the benefit of eliminating important hypotheses regarding catalytic activity and proton pathways, but not lead to positive assignment regarding how previously described proton transport activity is coupled to CO₂ hydration. Nevertheless, the overall study is a huge advance forward despite the enduring mechanistic mystery.

Specific Comments:

An important limitation of the mutagenesis studies is that the observed phenotypes cannot readily distinguish whether the mutations specifically impair catalytic turnover or proton translocation, as opposed to more general effects on protein stability or complex assembly. Without direct evidence for the proper accumulation of the mutant complexes, the functional consequences remain ambiguous. The application of immunodetection methods—such as antibodies against DabA2 or DabB2 or tags would strengthen the interpretation by confirming

that the mutant proteins are expressed and assembled at levels comparable to wild type, thereby ruling out destabilization as the primary explanation for the loss of activity.

Response:

We totally agree with Reviewer #3. Indeed, our site-directed mutagenesis experiments only inform the importance of amino acid residues, as reflected by the bacterial growth, rather than their specific function. To probe into the effect of point mutations on protein expression, we now performed additional LC-MS analyses (See Supplementary Fig. 11). Our findings suggest that the amino acid substitutions have minimal impact on the protein expression levels, therefore the observed phenotype is unlikely due to impaired protein expression.

The ATR-FTIR experiments provide useful supportive evidence that Dab2 interacts directly with CO₂, consistent with the cryo-EM densities. However, the data do not unambiguously prove supernumerary binding sites, nor do they definitively establish the catalytic vs. non-catalytic nature of hydration. The comparisons to BSA and ECCA are helpful but not definitive controls. The conclusion that Dab2 binds CO₂ “independent of membrane potential” is particularly weak, as the assay lacks physiological coupling conditions. Stronger biochemical validation (e.g., isotope-labeled turnover assays, PMF-dependent reconstitution) will be needed to confirm the functional significance of the FTIR observations.

Response:

The reviewer is correct, the FTIR data do not allow specifying the CO₂ binding site. However, in combination with the structural data the FTIR spectra provide strong support for supernumerary binding sites: as discussed in the main text, the increased band intensity suggests a high concentration of CO₂. Additionally, the 4–5 cm⁻¹ down-shift indicates an environment with limited access of water, e.g., a hydrophobic protein channel (also see Supplementary Fig. 14).

We agree that a catalytic CO₂ hydration activity of Dab2 cannot be ruled-out completely, based on the FTIR data alone. This is mentioned in the main text now. However, the similarity between Dab2 and BSA clearly points in the direction of “background” or un-catalyzed CO₂ hydration, as it is common in alkaline aqueous solution. We hope that the kinetic comparison in new Fig. 3c (see our comment to reviewer 1) makes this clearer now.

The interactions between CO₂ and purified proteins (as shown by ATR-FTIR experiments) suggests CO₂ binding do not require membrane potential. Nevertheless, it is unclear if PMF modulates the CO₂ binding under physiological conditions. To clarify, we refined our interpretation on line 320: “...This indicated that membrane potential is not essential for CO₂-binding...”. Thank you for suggesting biochemical experiments. Indeed, we are highly interested in the biochemical characterization of Dab2 but unfortunately this is beyond our current capacity. We considered Reviewer #3 suggestions as a probable follow-up study.

The hypothesis that the absence of a glutamine or histidine to stabilize, the transition state in the DAB protein is a quite reasonable explanation for the absence of catalytic activity in the isolated protein. However, in terms of scholarship, a better citation evidence for the role of the punitive transition states stabilizing, glutamine, or histamine that is absent in the DAB active site needs to be presented. The authors need to cite, e.g. a mutational study of that residue

rather than the structural studies if they want to make the argument that this residue is indeed crucial for catalytic activity in the canonical enzymes since the structural studies hypothesize the activity, but do not test it by mutagenesis.

Response:

Thank you for pointing this out. We have now included relevant references (ref. 31, 32).

Minor corrections:

Figure 2b: right panel—amino acids are mislabeled

Response:

We apologize for the mistake. We have now corrected the AA labels.

Reviewer #1 (Remarks to the Author):

The authors have taken all of the reviewers' suggestions and comments into account in the revised version of the manuscript. The current version of the manuscript can be accepted, with only a few minor points remaining to be addressed:

L. 75: The term “protonmotive force” is a technical term, whereby the first two words are written together.

L. 162: Delete ‘ref.’

L. 197: No activities are shown but growth curves: replace to ‘... had no significant impact on bacterial growth yield ...’

L. 307: Delete ‘ref.’

L. 1208: replace ‘explementary’ by ‘exemplary’

Response:

Thank you for the comment. We have now revised accordingly.

Reviewer #2 (Remarks to the Author):

I appreciate the authors' efforts to address this reviewer's concerns. I am satisfied with the responses to all comments I raised and the changes to the manuscript that have been made.

Reviewer #3 (Remarks to the Author):

The authors have adequately addressed my concerns regarding the manuscript.